# Seal milk oligosaccharides rival human milk complexity and exhibit functional dynamics during lactation

Chunsheng Jin [1], Jon Lundstrøm [2,3], Carmen R. Cori[4], Shih-Yun Guu[5], Alexander R. Bennett [6], Mirjam Dannborg[7,8,9], Patrick P. Pomeroy [10], Malcolm W. Kennedy [11], Johan Bengtsson-Palme [7,8,9], Rachel Hevey [4], Kay-Hooi Khoo[5] & Daniel Bojar [2,3] ✉

Milk oligosaccharides are crucial for neonatal development and health in mammals. Yet most milk research focuses on humans, or on domesticated mammals that are poor in milk oligosaccharide complexity. Here, we perform an exhaustive mass spectrometry-driven structural characterization of milk oligosaccharides in a wild mammal, female Atlantic grey seals (*Halichoerus grypus*), throughout their lactation period. Characterizing and quantifying 332 milk oligosaccharides, including 166 unreported structures, we reveal seals to rival human milk in complexity. We report seal free oligosaccharides to reach up to 28 monosaccharides in size. Paired glycomics and metabolomics time course analysis establishes a concerted regulatory process reshaping the seal milk glycome throughout lactation, similar to human milk. Functional analysis of the structures we here characterized reveals anti-biofilm effects and immunomodulatory functions of seal milk oligosaccharides. Our findings challenge long-held assumptions about milk complexity of non-human mammals and enable insights into the functional relevance of complex carbohydrates in milk.

Milk oligosaccharides (MOs), soluble glycans resulting from the elaboration of lactose in mammalian milk, are key contributors to infant development and health, humans being no exception[1]. The exact sequences of MOs in the resulting milk glycome can have measurable impacts on protecting neonates from pathogens, via competitive inhibition or by nurturing the initial microbiome through selection for MO degraders such as *Bifidobacteria*[2], which is crucial for the development of a neonate's intestinal tract[3].

Critically, structural diversity in MOs, including features such as fucosylation, sialylation, and sulfation, directly determines their functional specificities, enabling tailored interactions with pathogens, immune cells, and commensal microbes across species[4]. An evolutionary tuning of the milk glycome across species[5–7] suggests that understudied mammals, particularly those with distinct ecological pressures, may harbor unique MO repertoires with uncharacterized bioactivities.

[1]Proteomics Core Facility at Sahlgrenska Academy, University of Gothenburg, Gothenburg, Sweden. [2]Department of Chemistry and Molecular Biology, University of Gothenburg, Gothenburg, Sweden. [3]Wallenberg Centre for Molecular and Translational Medicine, University of Gothenburg, Gothenburg, Sweden. [4]Department of Pharmaceutical Sciences, University of Basel, Basel, Switzerland. [5]Institute of Biological Chemistry, Academia Sinica, Taipei, Taiwan. [6]Department of Medical Biochemistry, Institute of Biomedicine, University of Gothenburg, Gothenburg, Sweden. [7]Division of Systems and Synthetic Biology, Department of Life Sciences, SciLifeLab, Chalmers University of Technology, Gothenburg, Sweden. [8]Department of Infectious Diseases, Institute of Biomedicine, The Sahlgrenska Academy, University of Gothenburg, Gothenburg, Sweden. [9]Centre for Antibiotic Resistance Research (CARe), Gothenburg, Sweden. [10]Sea Mammal Research Unit, School of Biology, University of St Andrews, St Andrews, UK. [11]Institute of Biodiversity, Animal Health & Comparative Medicine, Graham Kerr Building, College of Medical, Veterinary and Life Sciences, University of Glasgow, Glasgow, UK.
✉e-mail: daniel.bojar@gu.se

The current best estimate for the pan-mammalian milk glycome, accounting for structural ambiguities, is >650 known structures[6], of which ~100 have been recently discovered by our group[5]. A large portion of the remaining structures have been discovered in humans, due to a strong research focus on human breast milk over the last century. Currently, over 200 unique MO structures have been characterized in human milk[6], with the largest human MO reaching 18 monosaccharide building blocks[8], but it is important to note that any individual human milk sample contains far fewer identifiable structures.

One reason for our lacking knowledge in non-human MOs is that sample access can be a bottleneck, especially given that domesticated mammals (e.g., cows, goats, sheep), which would be readily accessible, have much lower oligosaccharide levels than their wild counterparts[7,9,10], further hampering MO discovery. We have noted previously that reported milk glycome diversities are essentially a function of the number of published studies on a given species[6] and expect the true biochemical diversity of non-human milk to be far larger than currently assumed. This is especially relevant, as it is known that MOs also fulfil crucial roles in non-human mammals, such as intestinal health and microbiome development[3]. In fact, it is key to point out that physiological roles of milk oligosaccharides may also differ between species due to their relative concentration differences, for instance with exceptionally high MO levels in human milk[11].

Marine mammals, in particular, represent an underexplored frontier: their unique evolutionary trajectories and extreme environments (e.g., high pathogen exposure, rapid postnatal development) likely drive the evolution of specialized MOs with potent protective or developmental roles. Further, they present a sampling challenge, contributing to the lack of studies in this direction. Our previous findings on dolphin MOs[5] support this hypothesis.

Seals are an especially promising MO reservoir because they do not rely on carbohydrates as an energy source in their milk[12], leading to a high ratio of oligosaccharides to lactose, which in turn facilitates the characterization of many unique structures. Thus, MOs of phocid seal species such as hooded seal (*Cystophora cristata*)[13], Arctic harbor seal (*Phoca vitulina vitulina*)[14], and bearded seal (*Erignathus barbatus*)[15] have been previously characterized, typically identifying fucosylated structures, which are often used as decoy receptors[4], as well as unusually large MOs in bearded and hooded seals[16]. Yet even in this group of reference species for rich milk glycomes, we lack a thorough characterization of (i) the full structural MO complexity and (ii) their change over time during lactation.

We here present the full milk glycome of the Atlantic grey seal (*Halichoerus grypus*), with hundreds of free milk oligosaccharides in a unique longitudinal dataset following the lactation period. To the best of our knowledge, our study presents the largest effort to longitudinally study milk glycans in a wild mammal. In a previous initial survey, only six short milk oligosaccharides from this species had been reported in metabolomics data[17,18]. Next to a great degree of structural diversity (increasing the number of all known MOs by >20%), as well as the hitherto largest characterized MOs, we unveil concerted changes in the glycome profile throughout lactation via time course analysis, as well as a multi-omics analysis with paired metabolomics data. We then engaged in functional experiments to show that some of the structures that are discovered here and that change during lactation have potent anti-biofilm and immunomodulatory functions. We envision that the extraordinary richness in milk oligosaccharides of the milk of *H. grypus* can serve as a model system to improve our understanding of lactation and the health impact of the milk glycome in all mammals, including humans.

## Results
### Seal milk harbors substantial glycan sequence diversity
To map the species diversity of milk oligosaccharides (MOs) in *H. grypus* (grey seal), we analyzed samples from five individuals at four timepoints during the lactation period (days 2, 7, 13, and 17/18/19 after birth), providing a total of 20 biological samples. It is important to note that the lactation period differs dramatically in true, phocid seals[19], with the four days of *C. cristata* being the shortest of any mammal. *H. grypus* lactates 17 days on average[20], making our four time-distributed samples a comprehensive dataset for the entire lactation period.

We measured all our samples via neutral/acidic fractionation, lactose depletion, and exoglycosidase digestion by porous graphitic carbon (PGC) electrospray ionization (ESI) MS/MS and MS$^n$ and complemented this with permethylation to target sulfated MOs[21] via liquid chromatography-mass spectrometry (LC-MS/MS) and matrix-assisted laser desorption/ionization-mass spectrometry (MALDI-MS) for a representative acidic fraction. Overall, this represented a plethora of mass spectrometry measurements, allowing us to measure and quantify 332 unique milk oligosaccharides in seal milk, of which we structurally characterized 240 (Supplementary Data 1, Supplementary Figs. 1, and 2). This single study thus makes *H. grypus* the species with the second-most characterized MOs, behind *Homo sapiens* (Fig. 1a). As expected from the evolutionary loss of the CMAH gene in pinnipeds[22], we detected no Neu5Gc-containing glycan in *H. grypus* milk. Throughout lactation, ~50% of the total abundance derived from fucosylated, non-sialylated, glycans, while up to 40% stemmed from sialofucosylated structures. Non-fucosylated, sialylated glycans remained low in abundance (1–4%). This preponderance of fucosylated structures was more reminiscent of human milk than of MOs from domesticated mammals, e.g., bovine MOs[1], yet also matched the characterized glycans in the closely related seal species *P. vitulina*[14].

With 166 of the 240 structures (69%) being entirely unknown (defined as being absent from comprehensive milk glycan databases[6,23]), we argue that seal milk presents an entirely different sequence distribution compared to human MOs. A biodiversity analysis of structural epitopes, glycan sizes, and glycan branching across all species with measured MOs in the glycowork database[6,23] indeed revealed that seal MOs exhibited the second-highest alpha diversity, just behind *H. sapiens* (Fig. 1b). Given the drastic difference in sampling (decades of studying thousands of human milk samples with diverse methods vs 20 samples measured in one study), we argue that *H. grypus* has the potential to eclipse human MOs in diversity and complexity, which would overturn the assumption of lower MO complexity in non-human species.

To that effect, we noted a range of unusual structures among our discoveries (Fig. 1c), including the presence of hitherto unknown or striking substructures/motifs, discussed further below. Especially noteworthy structures here included a sulfated version of the ubiquitous and, arguably most famous, MO 2′-fucosyllactose (2′-FL), Fucα1-2Gal6Sβ1-4Glc, which we have previously discovered in bottlenose dolphin milk[5]. Sulfation frequently acts as an enhancer and modulator for binding specific lectins[24], and we speculated that 6S-2′-FL could present a more potent form of 2′FL for its typical anti-viral functions[4,25]. To that effect, we analyzed experimental glycan array data of 2′-FL and 6S-2′-FL and indeed uncovered both coronavirus as well as influenza virus proteins with strongly enhanced binding to 6S-2′-FL compared to 2′-FL (Supplementary Fig. 3a and Supplementary Data 2), supporting our hypothesis.

Additionally, we report the discovery of giant MOs in the milk of *H. grypus*, with several highly branched structures reaching 28 monosaccharides in length, such as Neu5Ac$_6$Hex$_{12}$HexNAc$_{10}$ (Supplementary Fig. 4 and Supplementary Data 3). This was substantially longer than the currently largest known human MO of size 18[8], and is among the largest mammalian non-polysaccharide glycans of any type (Supplementary Data 4), possibly due to the reliance on type-2 LacNAc repeats in seal MOs that can be extended, in contrast to human type-1 LacNAc[26]. We note that the scaffold of these mega-MOs allows for multivalent presentation of classic MO epitopes (Supplementary Fig. 4), such as sialylation and/or fucosylation, and thus could make

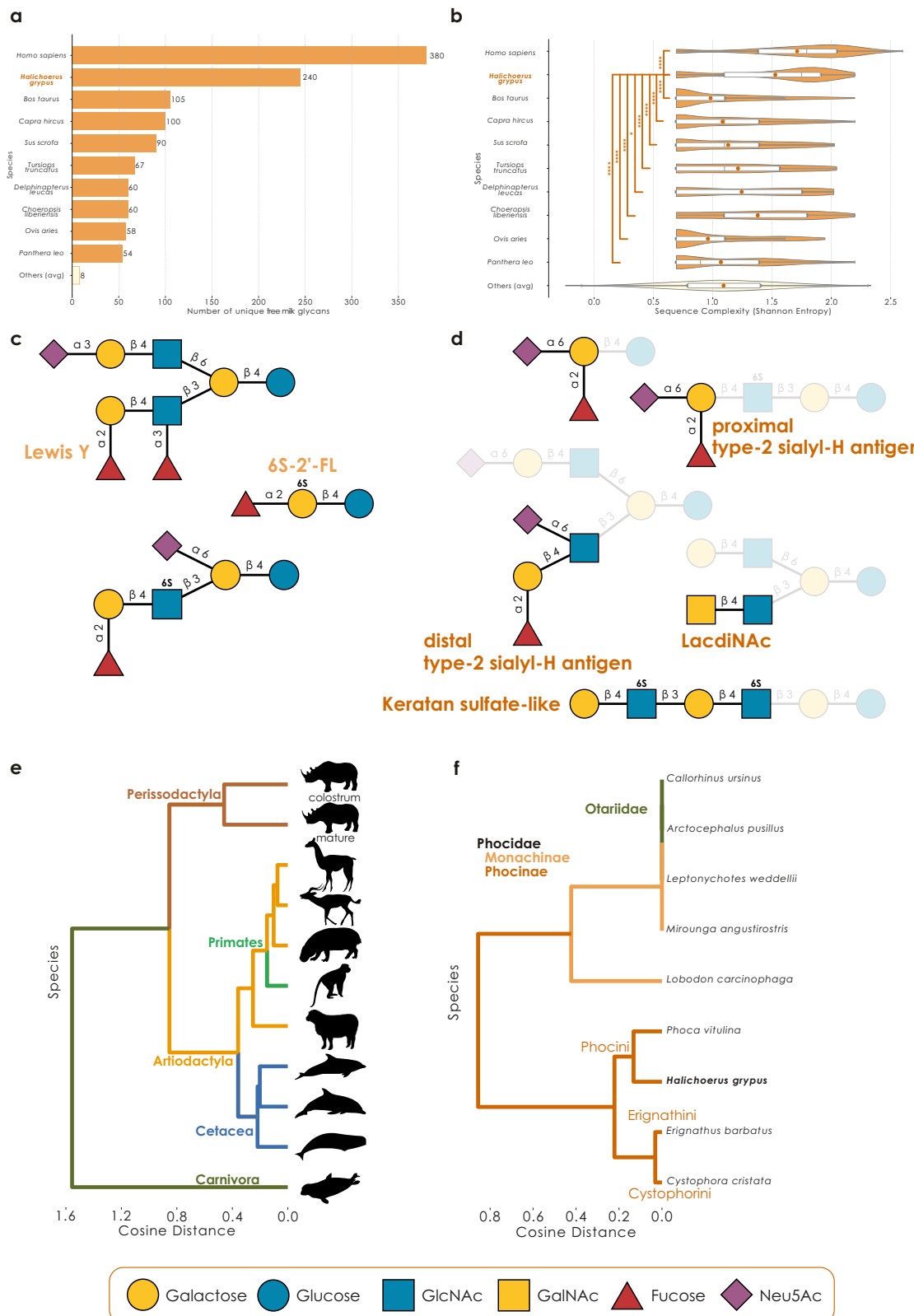

these molecules highly potent as soluble receptor decoys or signaling molecules themselves. As the presence of giant MOs has also been reported in the milk of hooded and bearded seals[16], with detected compositions of up to 21 monosaccharides, we suggest that this might be a shared trait of seal milk. We note that previous work has found such structures to be mainly fucosylated, whereas we here report both fucosylation and sialylation on giant MOs.

*H. grypus* uses lactose (Galβ1-4Glc) as a core, which is further elongated with multiple Galβ1-4GlcNAc (*N*-acetyllactosamine, LacNAc, type 2), namely poly-LacNAc, which is in contrast to human MOs that exhibit high levels of type 1 LacNAc (Galβ1-3GlcNAc), which select for *Bifidobacterium* species for an initial microbiome[26]. These poly-LacNAc chains serve as linear and extended scaffolds for diverse modifications, such as fucosylation, sialylation, and sulfation. Regarding noteworthy

**Fig. 1 | Grey seal milk exhibits a complex but characteristic glycome. a** Species with the most known free milk oligosaccharides. We displayed the 10 species with the most structurally characterized MOs, as well as the average of the remaining species, via a bar graph. **b** Alpha diversity (Shannon entropy) of structural epitopes, glycan sizes, and glycan branching across species. Data are shown as overlaid violin plots and boxplots. Lines in the boxplot indicate the median and dots the mean. The edges of the box describe the interquartile range (25% to 75%) and whiskers extend this to maximally 1.5× the interquartile range. Statistical comparisons between *H. grypus* and the other species have been performed as two-tailed Mann Whitney U-tests with a Benjamini–Hochberg correction. $n$ (top to bottom) = 380, 240, 105, 100, 90, 67, 60, 60, 58, 54, 1307. $^*p < 0.05$, $^{***}p < 0.001$, $^{****}p < 0.0001$. **c** Representative structures identified in the milk of *H. grypus*, chosen because of

them exhibiting unusual motifs. **d** Additional structural motifs identified in the milk of *H. grypus*. **e** Phylogenetic tree of free milk glycomes. Pairwise cosine distances of motif abundances for all species with available comprehensive and quantitative milk glycomics[5] were used to create a dendrogram via UPGMA. Animal illustrations were taken from https://www.phylopic.org/. Images for *Tursiops truncatus* and *Delphinapterus leucas* were originally published by Chris huh. PhyloPic.org (2012); released under a Creative Commons Attribution-ShareAlike 3.0 Unported license (https://creativecommons.org/licenses/by-sa/3.0/). **f** Phylogenetic tree of seal milk glycomes. Dendrograms were constructed similarly to (**e**), yet only using the presence/absence of terminal motifs in all known seal milk glycans instead. All glycans in this article are drawn with GlycoDraw[73] and comply with the Symbol Nomenclature For Glycans (SNFG).

motifs/substructures in seal MOs, we first note an abundance of known structural motifs, such as the Sialyl-Lewis X, Lewis Y, type-2 H antigen, B antigen, Galili antigen, I antigen, and i antigen motifs (Fig. 1c and Supplementary Data 1). Some of these, such as type-2 H antigens, are also similar to the MOs of other carnivores such as raccoons[27]. Of note, some common MOs, such as Galβ1-4GlcNAcβ1-6(Galβ1-3)Galβ1-4Glc (*novo*-LNP-I)[1], were not detected in seal milk. Next, tying into our recent discovery of the LacdiNAc motif (GalNAcβ1-4GlcNAc) as an additional yet common MO building block[5], we report here that *H. grypus* milk also exhibits LacdiNAc-containing MOs (Fig. 1d and Supplementary Fig. 1d), with 10 herein characterized seal MOs carrying this substructure (five of them not yet reported sequences; Supplementary Data 5). We further note one previously unknown structure with a sialylated LacdiNAc motif here as well.

Our in-depth mass spectrometry workflows also revealed the presence of two additional terminal motifs, Fucα1-2(Neu5Acα2-6)Galβ1-4GlcNAc/Glc and Fucα1-2Galβ1-4(Neu5Acα2-6)GlcNAc (Fig. 1d and Supplementary Fig. 1a, e), which, due to their biochemical provenance, we term proximal and distal type-2 sialyl-H antigen, respectively, referring to the glycan epitope that comprises blood group O, Fucα1-2Galβ1-4GlcNAc. Using the comprehensive glycan database within glycowork[23], we find that the proximal type-2 sialyl-H antigen has only been reported once so far, in glycosphingolipids of human colon adenocarcinoma[28], yet never in milk oligosaccharides. On the other hand, the distal type-2 sialyl-H antigen has only been described in glycosphingolipids of acute myeloid leukemia[29] and in MOs of another seal species[14], *P. vitulina*, raising the possibility (discussed below) of this as a more general seal motif. Both the H antigen and sialic acid moieties are potent anti-pathogen effectors in MOs[30], and we speculated that their combination in the same molecules could enhance their potency.

Analyzing the curated glycan array binding data stored in the glycowork library (Supplementary Fig. 3b and Supplementary Data 2), we indeed found that proximal sialylation changed the binding behavior of the type-2 H antigen (e.g., abrogating LEL binding, leaving UEA-I binding unaffected, and increasing AAL binding). In the case of distal sialylation, we again found increased AAL binding, as well as Siglec-1 and Siglec-15 binding (Supplementary Fig. 3c and Supplementary Data 2), supporting potential biological roles for these discovered motifs in being more potent immunomodulators.

Sulfation is a very common modification in seal MOs. Sulfated milk oligosaccharides have been reported as a minor constituent of HMOs, for instance in internal 6-sulfo Lewis X[6,31]. In our previous study, various mono-sulfated MOs were also detected in other mammals, including marine mammals[5]. Here, we report the presence of keratan sulfate-like milk oligosaccharides (Fig. 1d and Supplementary Fig. 1c), which could play key roles in supporting mucosal barrier function by mimicking host tissue glycosaminoglycans[32], fostering a glycosaminoglycan (GAG)-degrading microbiome, or potentially aiding in growth factor signaling[33]. Intriguingly, with the discovery of similar MOs with more repeat units, an additional keratan sulfate subtype[34], such as KS-IV, would need to be created for keratan sulfate originating

from a lactose core. In total, we noted 53 unique sulfated structures in *H. grypus* milk, 48 of which were not known before (Supplementary Figs. 5 and 6, and Supplementary Data 6), mostly via sulfation of the C6 of internal GlcNAc residues (Supplementary Fig. 1b, c), which is in contrast to the C3 sulfation of terminal Gal residues, reported in other seal species[15,16]. This, by far, made *H. grypus* the species with the most known sulfated MOs (compared to the 22 characterized sulfated MOs in *H. sapiens*[6]). This also further highlighted the promise of applying the relatively young technique of sulfoglycomics to more milk samples and indicates that the prevalence of sulfated MOs has been underestimated considerably thus far.

In general, non-sulfated seal MOs showed a deterministic pattern of extending sequences via branching and further decoration (Fig. 2), which then continued on to the measured giant MOs (Supplementary Fig. 4 and Supplementary Data 3). Next to modulating binding, as mentioned above, we thus speculated that sulfation—changing the electrostatic and steric environment of a monosaccharide—may also affect glycan extension. Specifically with the case of GlcNAc6S, we indeed observed that MOs containing sulfated GlcNAc, usually connected to a branchpoint galactose, were less branched than those containing unmodified GlcNAc at this position (Supplementary Fig. 5; Wilcoxon signed-rank test of extending sulfated/non-sulfated structures, controlled for their length: $p < 0.001$ that GlcNAc6S-modified structures are extended less often), leading us to the hypothesis that GlcNAc sulfation may be a mechanism in seal MOs to modulate and control branching, due to a change in substrate presentation to glycosyltransferases.

In previous work[5], we have shown that our detailed investigation of various milk glycomes resulted in well-comparable data and recapitulated DNA-based phylogenetic relationships between species. Since *H. grypus* is a member of Carnivora, a taxonomic order we have not investigated in our prior work, we were curious to probe whether this would be reflected in our glycan-based phylogenetic tree. Indeed, *H. grypus* (as the only representative of Carnivora in our dataset) clustered separately from the ungulate clade (Artiodactyla and Perissodactyla; Fig. 1e), particularly driven by the high levels of type 2 H-antigen and Poly-LacNAc in seal milk, combined with the absence of motifs such as Sd[a] (especially prevalent in Perissodactyla and Cetacea).

While our *H. grypus* dataset here presents the only quantitative seal MO dataset thus far, we were still interested in comparing the motif distributions of the various investigated seal species, to probe whether even finer taxonomic information can be found in their milk glycomes. For this, we used the absence/presence of motifs in identified milk glycans to calculate distances between seal milk glycomes, and construct a corresponding phylogenetic tree (Fig. 1f). Interestingly, this not only captured the main distinction within pinnipeds, of the families of eared seals (Otariidae) and earless, true seals (Phocidae) —diverging about 25 million years ago[35]—but even distinguished the Phocidae sub-families of Monachinae (34 chromosomes) and Phocinae (32 chromosomes). Pertinent to the Otariidae/Phocidae divergence, this is likely grounded in an alpha-lactalbumin mutation in the otariids that prevents them from forming elaborate MOs[19,36,37].

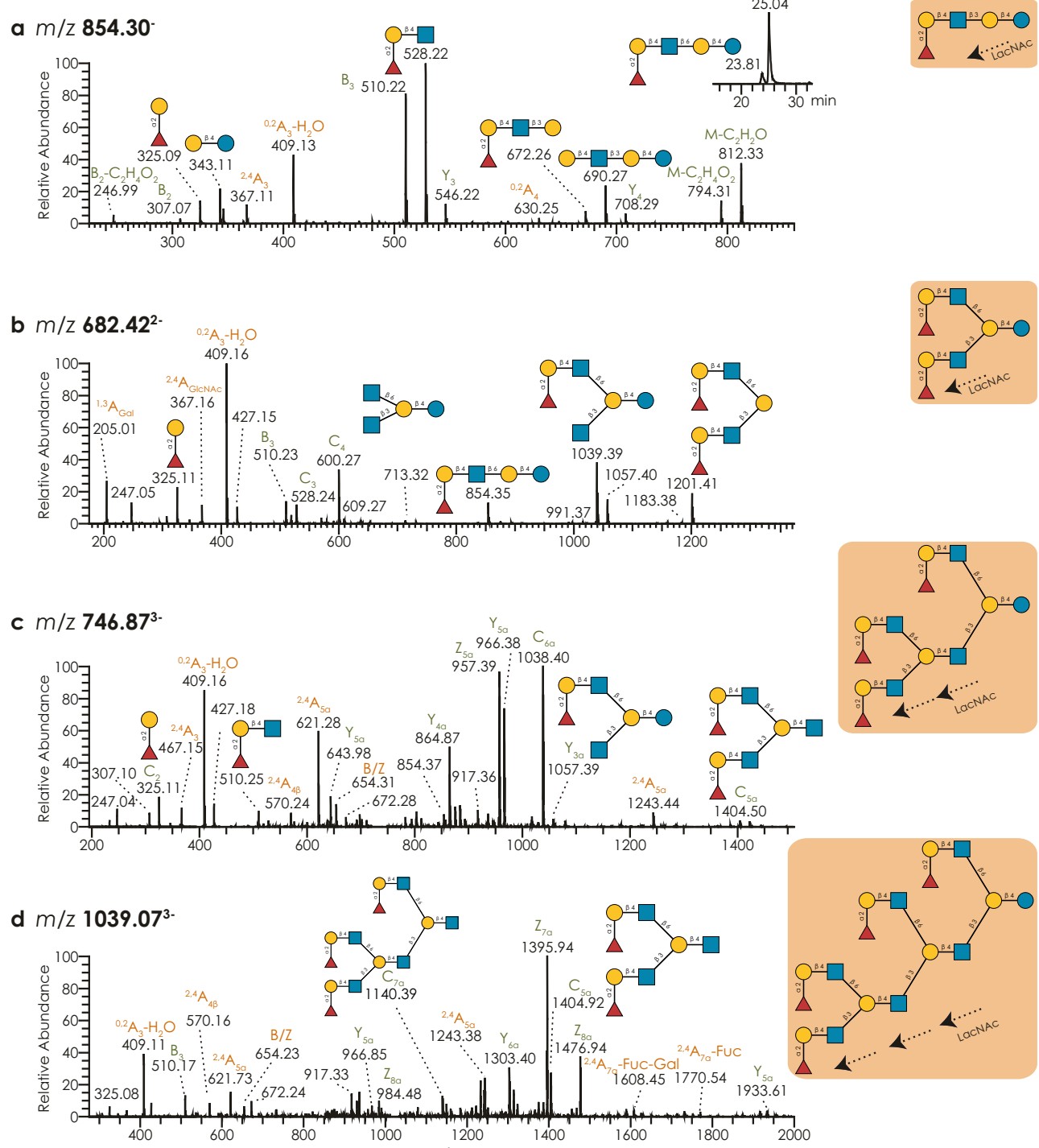

**Fig. 2 | Seal milk oligosaccharides are extended in a repeating pattern of branched poly-LacNAc.** Lactose is extended with one or multiple LacNAc units (**a**–**d**) and further branched with one or more LacNAc units (**b**–**d**). For the examples of neutral, fucosylated MO structures at *m/z* 854.3 (**a**), 682.42 (**b**), 746.87 (**c**), and 1039.07 (**d**), we show representative and annotated MS² spectra, as well as the full determined sequence. Further, the chromatogram in **a** shows the separation of the two GlcNAc linkage isomers in retention time. MS/MS spectra of I-antigen containing structures are rich in B/C ions carrying this epitope (e.g., *m/z* 1201 in **b**, *m/z* 1038 and 1404 in **c**, and *m/z* 1140 and 1404 in **d**), 2,4A cleavage of GlcNAc adjacent to the branched Gal residue (e.g., *m/z* 621 in **c** and *m/z* 1770 in **d**), and D ions which indicate the size of the C6 branch caused by double cleavages of branched Gal residues (e.g., *m/z* 654 in **c** and **d**). All fragments in this work are provided in the Domon–Costello nomenclature[74].

Among seals with MOs that have been investigated, the closest neighbor to our *H. grypus* data in this clustering was *P. vitulina* seals. As both these species belong to the Phocinae sub-group of Phocini, we conclude that even fine-grained evolutionary information is reflected in the milk glycomes of these species. This clustering of *H. grypus* and

*P. vitulina* was in part driven by their shared expression of the distal type-2 sialyl-H antigen, one of the unusual structures mentioned above (Fig. 1d). Overall, our glycan-derived seal taxonomy matched genomic phylogenies of this clade[38], especially considering that we lacked quantitative glycan data for most of these species. We thus conclude

that milk glycans are exquisite repositories of evolutionary information, due to their adaptation to specific environmental niches.

## The seal milk glycome is changing throughout lactation
Next, we wanted to make use of the unique opportunity of having milk samples of the same seal individuals throughout their entire lactation period. For humans and cows, it has been shown repeatedly that the milk glycome is dynamic and changes over time[39,40], to fit the changing needs of the infant. It is not known whether this is the case for seals, especially given their relatively brief lactation period, so we next investigated whether our milk glycomes would cluster by time point (Supplementary Data 7). For this, we used hierarchical clustering of CLR-transformed relative glycan abundances (Supplementary Data 8), since glycomics data are compositional data which results in, otherwise unaccounted for data dependencies[41]. We observed a strong clustering of samples by their time point (Fig. 3a), substantiated by high clustering metrics, which indicated that the seal milk glycome undergoes a concerted change during lactation that is conserved across individuals, which is especially remarkable since the lactation period in grey seals is much shorter compared to humans or cows.

We further observed that several clusters of MOs exhibited distinct temporal dynamics (Fig. 3a), which we further investigated by analyzing the total glycan abundance in each cluster (the four top-level row clusters from the hierarchical clustering) over time (Fig. 3b). This analysis led to the conclusion that there were three temporally important clusters (as well as a fourth group of glycans with very low and sporadic expression): a cluster of glycans exclusively expressed in early milk (Early), another cluster with relatively stably expressed MOs (Stable), and a last glycan cluster with late MOs that were not found in early milk and increased in abundance over the later stages of lactation (Late).

Reasoning that these different MO clusters fulfilled different functions in seal milk, similar to what is documented for human milk[40], we next set out to investigate which functional MO motifs distinguished each cluster from the others. We caution that, in addition, each cluster also contained many other motifs, yet these were not necessarily characteristic of that cluster. Overall, an analysis of variance (ANOVA)-based workflow of motif-level abundances allowed us to show that early MOs exhibited high levels of the alpha-Gal motif (Galα1-3Gal), while late glycans were enriched for the Lewis Y antigen and sulfated MOs, especially the mentioned keratan sulfate-like structures (Fig. 3c). We note that sulfated MOs in general have also been reported to increase in later lactation stages in cow milk[42]. Lastly, the stable cluster was characterized by motifs such as the type 2 H-antigen and internal LacdiNAc structures. This analysis then confirmed our initial hypothesis that the different temporal clusters exhibited a unique array of functional moieties in their MOs, predisposing them for fulfilling different niche functions during pup development.

While analyzing these different temporal dynamics of the clusters can yield insights into the regulation of the lactation cascade, we next wanted to analyze how the milk glycome as a whole changed during lactation. With the example of fucosylation (Supplementary Fig. 7), we revealed that some of its dynamics can be complex to disentangle, such as with a general decrease in fucosylated glycans during lactation, which was mainly driven by decreasing Fucα1-2Gal-containing glycans (important for blood group epitopes), even though Fucα1-3GlcNAc-containing glycans (important for Lewis antigen epitopes) exhibited the opposite trend, which is similar to the dynamics of human MOs[40].

Taking this analysis method to some of our herein discovered motifs, we found, in accordance with our cluster analysis (Fig. 3c), that keratan sulfate-like MOs exhibited an increasing abundance even when considering the entire milk glycome (Fig. 3d). Further, we noted a general decrease in the abundance of LacdiNAc-terminated MOs during lactation (Fig. 3e), indicating that their levels seemed to be highest in early milk. This decrease in LacdiNAc-containing MOs during lactation resembled a similar decrease in LacdiNAc which has been reported in protein-linked glycans in cow milk[43], raising interesting possibilities of cross-connections between different glycan types in milk during lactation.

Lastly, we returned to our milk glycome clustering (Fig. 3a) to make another observation: Next to temporal clusters of MOs, it also seemed to us that the milk glycome of *H. grypus* became generally more diverse in later lactation stages. We formally analyzed this via three different alpha diversity indices of the milk glycomes and conclusively confirmed this observation (Supplementary Fig. 8a–c). Next, we wanted to make sure that the clustering of the samples on our heatmap was not merely due to this increase in diversity and engaged in an ANOSIM analysis of the beta diversities of our milk glycomes (Supplementary Fig. 8d), which confirmed that the glycan sequence content also changed throughout lactation, additionally supported by our earlier analyzes. Overall, we report that the seal milk glycome (i) becomes more diverse throughout lactation, (ii) changes its repertoire of available functional groups, and (iii) exhibits clusters of glycans with concerted changes, e.g., only present during the early phase, that hint at a regulated and conserved process.

## The changing seal milk metabolome is reflected in the glycome
Since the exact same seal milk samples have been used for metabolomics measurements in an earlier study[18], we next set out to investigate whether we could find links between the changing milk metabolome (Supplementary Data 9) and the milk glycome in *H. grypus*, given that glycosylation is metabolically regulated[44]. Using our established approach of cross-correlating CLR-transformed systems biology datasets[41], we did indeed find many significant correlations between glycan substructures/motifs and milk metabolites throughout the lactation period (Fig. 4a).

Given the high degree of overlap between shared glycan substructures, as well as biosynthetically related metabolites, we then refined this by calculating regularized partial correlations, correcting for "bystander" correlations and enriching for more direct effects (Supplementary Data 10). We noticed strong positive correlations (Spearman's ρ of 0.7–0.9) between several MO substructures and membrane lipids (phosphatidylcholines, phosphatidylethanolamines, phosphatidylinositols, sphingolipids) as well as fatty acid components (acylcarnitines), which could indicate the coordinated delivery of energy via milk fat and developmental/protective factors via the MOs.

Specific regularized partial correlations, for instance, included a strong negative correlation of sialylated MOs (Neu5Acα2-3Gal) and the anti-inflammatory[45] eicosanoid prostaglandin A$_2$ (Fig. 4b), with the latter increasing in later stages of lactation. Additionally, we noted a strong positive correlation between sulfated glycans (Galβ1-4GlcNAc6S) and 2-oxophytanate (Fig. 4c), a metabolite produced during the oxidation of phytanic acid, which is derived from a fish diet[46]. This could be a product from lipid catabolism, since grey seals are capital breeders (building body fat for the lactation period) and fast for the entire lactation period, in contrast to income breeders (using current food intake to fuel lactation). Increasing lipid catabolism is known to increase oxidative stress, whereas sulfated glycans have been linked to mitigating oxidative stress[47], creating an intriguing connection for future research into sulfated glycans as a protective factor here.

Given that the milk metabolome is also known to change over time[48], we went on to examine which systems biology modality, the milk glycome or metabolome, carried clearer information about the lactation stage. Similar to our previous clustering analysis (Fig. 3a), we thus clustered samples by their glycome, metabolome, or both (Fig. 4d). Interestingly, this resulted in the glycome being the most informative modality for determining lactation stage (ARI: 0.539, NMI: 0.727), even superior to the combination of glycome and metabolome

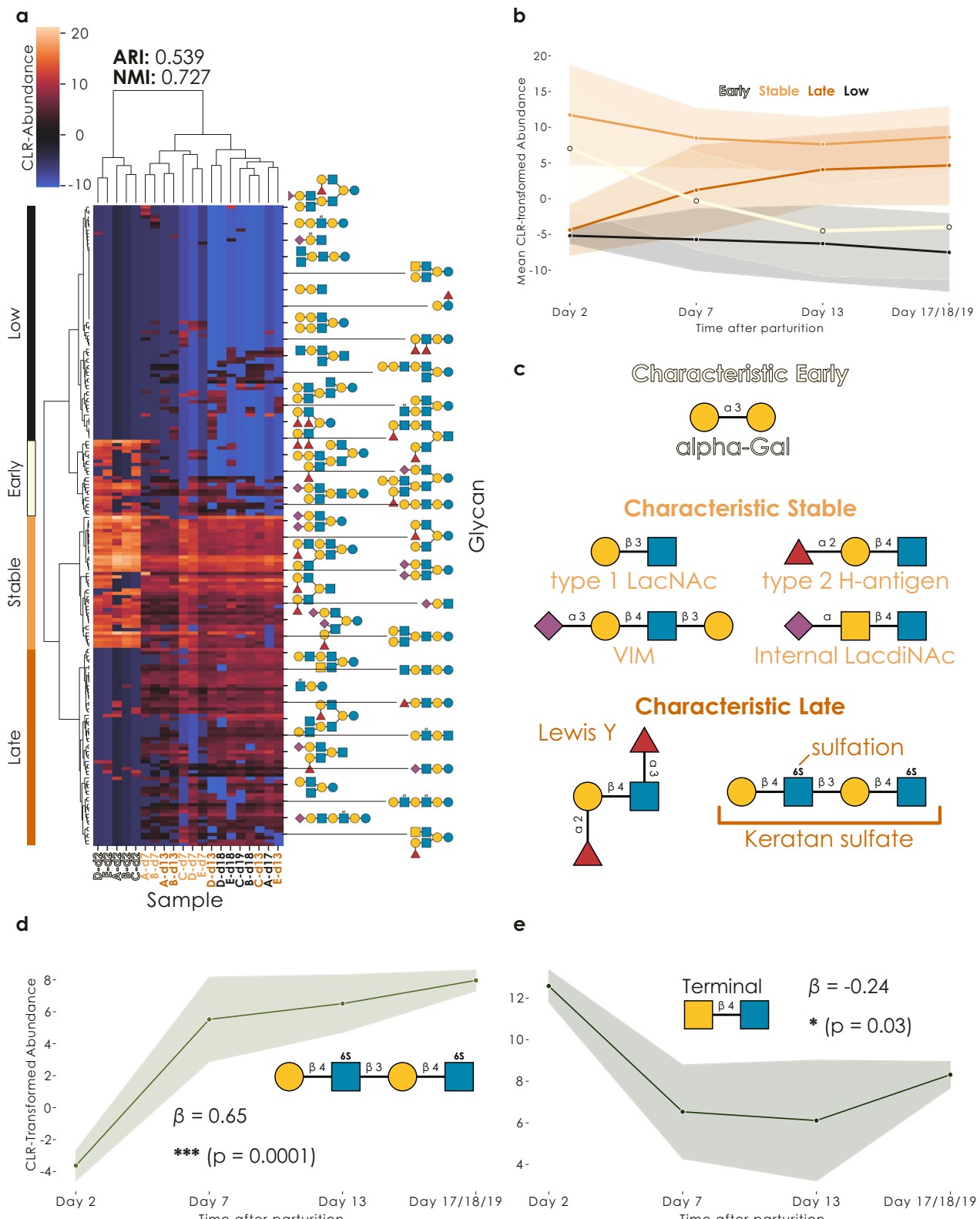

(ARI: 0.228, NMI: 0.429). We suspect that this latter result was due to the added noise by the metabolome, because PCA-driven denoising indeed improved the clustering by the combined features (ARI: 0.399, NMI: 0.550), yet these metrics still did not reach the performance of the milk glycome by itself, which could be explained by the metabolome clustering by individual animal, rather than lactation stage. We note that there is still a shared component of variation in both 'omics

layers that correlates similarly with lactation stage (Fig. 4e), explaining ~38% of variance, and loading highest on our keratan sulfate-like MOs, distal sialyl-H antigen, and fatty acids, respectively, indicating a shared physiological program.

Overall, this discrepancy in information content—despite the metabolome exhibiting ~5× the number of features of the glycome—clearly indicates the (i) physiological relevance and (ii) concerted

**Fig. 3 | The seal milk glycome is changing during lactation. a** Seal milk samples cluster by lactation time point. Using CLR-transformed glycomics data from five individuals (A-E) and four timepoints (d2, d7, d13, d17/18/19; total $N = 20$), we engaged in hierarchical clustering of samples via Ward's variance minimization algorithm. Representative glycans are shown for periodic rows. Successful clustering was assessed via the adjusted Rand index (ARI; ranging from −1 to 1) and normalized mutual information (NMI; ranging from 0 to 1). The four main glycan clusters resulting from row-wise clustering are labeled. **b** Distinct clusters of glycans change abundance in a concerted manner during lactation. Data are shown as the abundance means of all glycans in the cluster for that timepoint, connected by a line plot with a 95% confidence band, shaded by cluster identity. **c** Specific motifs characterize the temporally regulated glycans during lactation. For each cluster (Early, Stable, Late), we used the *quantify_motifs* function in glycowork (v1.5)[23] to obtain cluster-specific motif abundances and then determined which motifs were

most cluster-specific via an ANOVA, using the *get_glycanova* function from glycowork. Representative, cluster-specific motifs are shown for each cluster via their SNFG depiction. All shown motifs are significantly different from all other clusters ($p < 0.05$), determined via two-tailed Tukey's Honestly Significant Difference (HSD) post-hoc tests, followed by a two-stage Benjamini−Hochberg correction for multiple testing. Terminal epitopes change consistently through lactation. We used the *get_time_series* function of glycowork (v1.5) to analyze the expression of keratan sulfate-like (**d**) and terminal LacdiNAc-containing glycans (**e**) throughout the lactation period. Shown are line plots of the CLR-transformed motif abundances, with a 95% confidence band around the line indicating the mean value, as well as the regression coefficient ($\beta$) and its significance (via a two-tailed, one-way ANOVA followed by Benjamini−Hochberg correction for multiple testing), from fitting a degree 1 polynomial function to the time series. *$p < 0.05$, **$p < 0.001$.

regulation of the MOs during lactation, as well as the substantial information richness of glycans in general.

### Changing structures in seal milk glycome exhibit immunomodulatory and anti-biofilm properties

Based on the prominence of LacdiNAc-containing structures in our seal milk samples (Fig. 1) and that we find this motif to be conserved in milk oligosaccharides[5], we decided to more closely investigate functional properties of this class of molecules. MOs have known and potent effects on modulating immune cell activity beyond the gut itself that could enhance a neonate's systemic immune defense and stimulate the maturation of their immune system, and influential MOs are often conserved across species[5,49]. In the absence of an available *H. grypus* immune cell line, we thus tested the impact of LacdiNAc on human macrophages.

As a baseline for comparisons, we first differentiated THP−1 monocyte cells into naïve macrophage-like cells (M0 macrophages). These were then activated with LPS for classical (M1) activation or IL-4/IL−13 for alternative (M2) activation, leading to distinct cytokine profiles across macrophage populations (Fig. 5a). Next, we co-treated naïve, M1-, and M2-stimulated macrophages with LacdiNAc (GalNAcβ1-4GlcNAc) and the very closely related LacNAc (Galβ1-4GlcNAc) moiety. In addition to these disaccharides, we synthesized a full LacdiNAc-containing MO (LdiNnT, lacto-*N,N*-neotetraose), for comparison with the conserved MO LNnT (lacto-*N*-neotetraose), differing by only one *N*-acetyl group.

With the possible exception of LNnT, which was produced by fermentation and thus may contain trace endotoxins, treatment with (physiologically relevant concentrations[50] of) LacNAc/LacdiNAc structures had little to no effect on cytokine production in naïve macrophages (Fig. 5b and Supplementary Fig. 9). In contrast, we observed numerous significant immunomodulatory effects of LacdiNAc-containing structures (both *N,N*-acetyllactosamine and lacto-*N,N*-neotetraose) but not LacNAc-containing structures (*N*-acetyllactosamine and lacto-*N*-neotetraose) in activated macrophages. Specifically, in M1-polarized macrophages, LacdiNAc upregulated CCL17 and IL−10 (Fig. 5c, e), while in M2-polarized macrophages, it downregulated IL-12p40, IL-12p70, IL-1β, and IL-23 (Fig. 5d, f). All results can also be found in Supplementary Data 11.

Previously, LacdiNAc has been proposed as a parasite-associated pattern and as a ligand for galectin-3 on macrophages[51]. We contend here, however, that galectin-3 cannot be viewed as the only LacdiNAc receptor responsible for our observed immunomodulation because: (i) the above-cited work did specifically not identify galectin-3-mediated effects in the herein used THP−1 cells; and (ii) galectin-3 in fact bound LacNAc with at least similar affinity to LacdiNAc[51], while we observed significant differences in the effects of LacNAc vs LacdiNAc in our assay. We thus hypothesize that another LacdiNAc receptor exists on THP-1 cells that contributes to the immunomodulatory effect of this glycan moiety. Macrophage Galactose-type Lectin, which is highly

expressed in macrophages and further upregulated under M2-polarizing conditions[52], specifically recognizes GalNAc-terminated structures, including Tn and LacdiNAc[53]. It could therefore be a candidate receptor mediating the LdiNT-specific effects observed here, although further research is needed to confirm this.

Next to immunomodulatory effects, an emerging property of MOs in recent years has been the inhibition of biofilm formation[54–56], with important implications for antibiotic resistance development, which we can confirm here, for instance, with the well-characterized 6'-sialyl-lactose (Supplementary Fig. 10). Since LacdiNAc-containing glycans have never been assessed in this regard to the best of our knowledge, we continued our comparison of LacNAc and LacdiNAc with respect to their action on biofilms. Assessing three pathologically relevant bacterial strains (*Klebsiella pneumoniae* KP1, *Streptococcus agalactiae* CCUG 4208T, *Staphylococcus aureus* CCUG 1800T), we report that LacdiNAc−but not LacNAc−inhibited biofilm formation in *S. agalactiae* and *S. aureus*, while both glycan moieties inhibited *K. pneumoniae* biofilm formation (Fig. 5g), without affecting bacterial growth (Supplementary Fig. 11). Importantly for potential resistance development, none of the added glycan moieties affected bacterial growth (Supplementary Fig. 11), potentially indicating either a signaling-mediated effect of the added glycan moieties or an inhibition of biofilm-associated lectins.

These encouraging findings implied that (i) more anti-pathogenic molecules are still available in mammalian milk for biomining purposes and (ii) minute chemical differences, such as between LacNAc and LacdiNAc, coupled with pronounced biological differences, speak to a specific recognition mechanism mediating this effect, which could eventually be therapeutically targeted. It is also interesting to note here that mucin *O*-glycans, which can contain similar epitopes, have been speculated to modulate *S. aureus* virulence and adhesion in vivo[57,58]. Therefore, we continued this line of research by probing the anti-biofilm properties of the recently discovered glucuronyl-lactose[5] as another recently discovered MO, which then demonstrated promising anti-biofilm properties in *K. pneumoniae* and *S. aureus* that could not be observed with free glucuronic acid (Supplementary Fig. 12). We thus conclude that anti-biofilm properties could be common in yet-to-be-discovered MOs and deem this a promising reservoir for mining anti-pathogenic compounds.

## Discussion

With an in-depth case study of the multifaceted and longitudinal milk glycome of the Atlantic grey seal, *H. grypus*, we here show that human-level complexity of milk oligosaccharide biochemistry and regulation can be found elsewhere in the animal kingdom, revising currently held paradigms of the exceptional state of human breast milk. While human milk may still be exceptional regarding MO quantity[11], we show with our work that biochemical diversity can also reach high levels in non-human mammals. Further, the heterogeneity we observe across our individuals (Fig. 3a) should motivate both exploring the milk of more

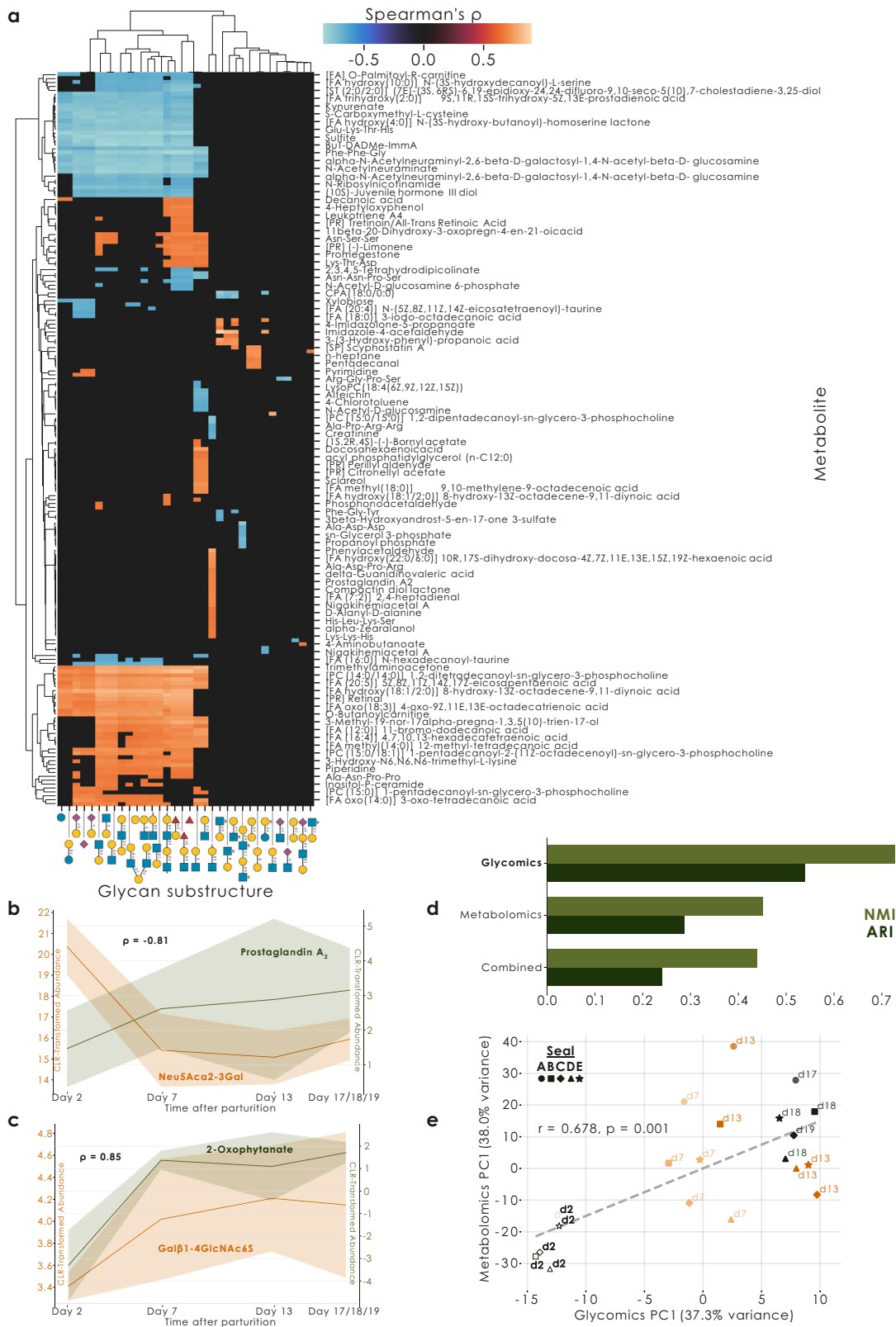

individuals per species and elucidating the genetic basis for this observed polymorphism. We also note that we here, yet again, extend the number of all known MO structures by >20%, demonstrating how much remains to be learned about glycan biosynthesis and biodiversity. Finally, the highly potent functional properties of structural motifs in these herein discovered MOs, if nothing else, should motivate the further exploration of the MO repertoires of more mammals.

As noted previously[5,12], aquatic and semi-aquatic mammals (such as *H. grypus*) exhibit a pronounced MO complexity, and we envision that future efforts to map the pan-mammalian milk glycome will benefit most from investigating such species. While we present pioneering evidence that motifs such as sialyl-Lewis X, Lewis Y, B antigen, and the Galili antigen can also be present in seal milk, some of our motifs, such as type-2 H antigen, Lewis X, B antigen, or the Galili

**Fig. 4 | The changing milk metabolome is reflected in the glycome. a** Seal milk glycan substructures and metabolites correlate throughout the lactation period. We used the *get_SparCC* function from glycowork (v1.5) for a cross-correlation analysis of CLR-transformed glycomics and metabolomics data. Shown is a hierarchical clustering of the resulting Spearman's ρ correlation coefficients, with only significant correlation coefficients shown ($p < 0.05$ of two-tailed $t$-tests, corrected for multiple testing by a two-stage Benjamini–Hochberg procedure). **b, c** Features of the milk glycome and metabolome are strongly correlated during lactation. For the example of Neu5Acα2-3Gal/Prostaglandin A₂ (**b**) and Galβ1-4GlcNAc6S/2-Oxophytanate (**c**), we show their CLR-transformed abundance during the lactation period (line: mean, confidence band: 95% CI), as well as their regularized partial

correlation as Spearman's ρ. **d** Milk glycomes are more characteristic of lactation stage than milk metabolomes. Metrics (normalized mutual information, NMI; adjusted Rand index, ARI) are shown for clustering lactation timepoints by using CLR-transformed glycomics, metabolomics, or glycomics+metabolomics data. **e** Joint glycome-metabolome transitions mark lactation progression. Principal component analysis (PCA) was performed on CLR-transformed glycomics and metabolomics data. Correlation between glycomics PC1 (37.3% variance) and metabolomics PC1 (38.0% variance) is shown as Pearson's r, with its statistical significance assessed by a two-tailed $t$-test. Samples are colored by sampling day (d2-d19) and exhibit different shapes for each seal (A-E).

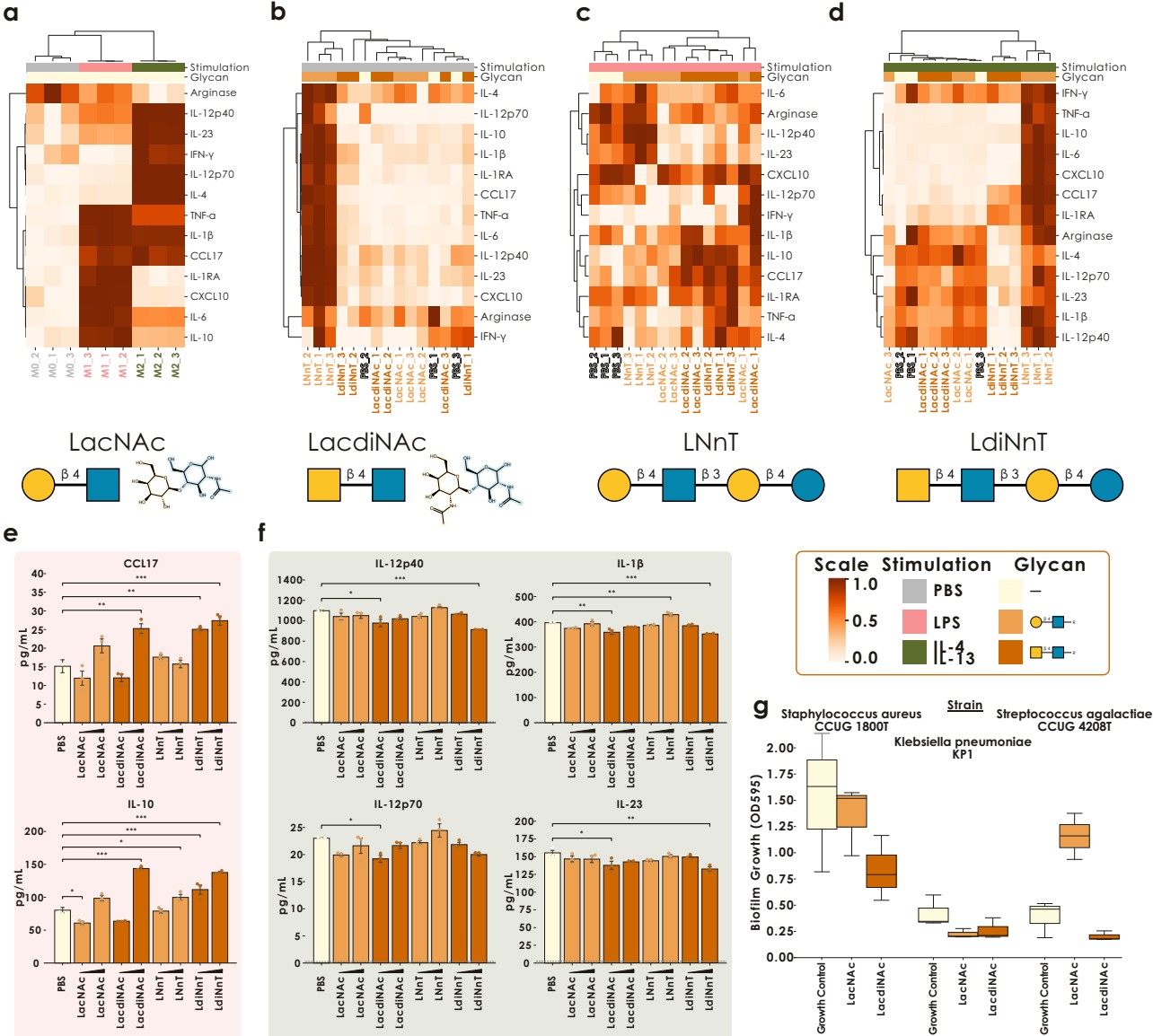

**Fig. 5 | LacdiNAc exhibits unique immunomodulatory and anti-biofilm effects. a** Cytokine profile of naïve (M0, PBS) vs M1-polarized (LPS) vs M2-polarized (IL-4/IL-13) macrophages. Cytokine profiles showing the immunomodulatory effects of LacNAc/LacdiNAc and LNnT/LdiNnT on M0- (**b**), M1- (**c**), and M2-polarized (**d**) macrophages. Row dimensions were normalized to 0-1 before clustering by correlation distance. **e** Quantification of CCL17 and IL-10 cytokine production upon 0.1–1 mM LacNAc/LacdiNAc and LNnT/LdiNnT treatment of M1-polarized macrophages. Error bars indicate standard deviation. $n = 3$ biological replicates **f** Quantification of IL-12p40, IL-12p70, IL-1β, and IL-23 cytokine production upon 0.1–1 mM LacNAc/LacdiNAc and LNnT/LdiNnT treatment of M2-polarized

macrophages. The dashed line indicates the limit of detection as determined by the standard curve of each analyte. Error bars indicate standard deviation. $n = 3$ biological replicates **g** Biofilm formation (measured as OD595 absorption, adjusted for blank medium controls) of three bacterial strains grown in the absence or presence of 1 mg/mL LacNAc or LacdiNAc. Lines in the boxplot indicate the median and dots the mean. The edges of the box describe the interquartile range (25% to 75%) and whiskers extend this to maximally 1.5× the interquartile range. $n = 3$ technical replicates. Significant differences were established via a one-way ANOVA with post-hoc Tukey's HSD tests. $^{***}p < 0.001$; $^{**}p < 0.01$; $^{*}p < 0.05$.

antigen have been identified in carnivore milk before[27,59,60]. Further, characteristics such as branching patterns might be conserved across Carnivora as other milk glycomes from this group also exhibit a poly-LacNAc branching pattern that is dominated by extension of the GlcNAcβ1-6 branch[27], although previous work on seal MOs has also reported substantial expansion on the GlcNAcβ1-3 branch[16]. In turn, other carnivore milk motifs, such as the A antigen[6], seem to be absent from the seal milk glycome. Our efforts here notwithstanding, the current rate of discovering >50% unreported sequences in each newly characterized mammal indicates that there is still much to be discovered in the biochemical diversity of milk oligosaccharides, especially in underexplored wild mammals, which could also shed light on biosynthetic constraints and yield functional molecules with biomedical relevance.

We especially note that the rather common properties of freshly characterized MO building blocks as precision probes for biomedically relevant applications such as immunomodulation or anti-biofilm action is reminiscent of the recently emerging consensus of mucin glycans as anti-virulence signaling molecules[61,62]. Particularly given their role as already soluble molecules, MOs are not only soluble decoys for viruses and bacteria but also prime signaling molecules with established functions, and given the partly shared sequence content with mucin O-glycans, we envision that there could be substantial synergy in studying these anti-pathogen signaling functions across mucin glycans and MOs. Our findings, such as increased binding of 6S-2'-FL to the dendritic cell immunoreceptor 1 compared to 2'-FL (Supplementary Fig. 3a), provide ample opportunities for starting this line of inquiry. Importantly, recent evidence suggests that both mixtures of mucin-derived glycans and MO structures exhibit enhanced or differential activities compared with the individual components in isolation[63,64], highlighting the need for further systematic investigation into the complex and interdependent regulatory mechanisms involved.

Deep quantitative glycomics datasets of non-human samples are still a rarity and—especially with (i) longitudinal data, (ii) milk, and (iii) stemming from several wild individuals such as in our work here—present a strong point of impact of our study and enable the kind of sophisticated analyzes that constitute the future of glycomics[65]. Crucially, this provides an exhaustive overview of available glycan epitopes in the free milk glycome. While this already facilitates analyzes across, e.g., neutral and acidic fractions of MOs, we envision that future efforts will also contribute the corresponding N- and O-glycans from proteins in these milk samples. This could, for instance, yield insights into total glycome vs glycan class-specific regulation of carbohydrate expression. We anticipate that more such multi-glycomics datasets, enabled by recent workflows[66], will yield a more exhaustive understanding of the regulation, dynamics, and functions of the milk glycome and its roles in physiology.

## Methods
### Sample processing
Milk samples were originally collected in a previous study[17] from a seal colony on the Isle of May, Scotland. Female subjects received weight-proportioned anesthesia using Zoletil 100® (Virbac, Bury St Edmunds, Suffolk, UK), after which intravenous oxytocin was administered to induce milk ejection, followed by intramuscular tetracycline as a preventive antibiotic measure. The oxytocin treatment consisted of a 1 mL intravenous dose (10 IU mL⁻¹ or 0.18 mg mL⁻¹; Oxytocin-S, Intervet UK). Given that post-parturition female grey seals in this population typically weighed -180 kg, the resulting oxytocin dosage was calculated at roughly 1 µg kg⁻¹.

Sample processing began with diluting 500 µL of breast milk with an identical volume of distilled water, followed by centrifugation at $4000 \times g$ for 30 min at 4 °C and a removal of the fat layer. We then added two volumes of cold 96% ethanol to one volume of skimmed milk, followed by overnight incubation at 4 °C to achieve protein precipitation. The sample underwent another centrifugation at $4000 \times g$ for 30 min at 4 °C, after which the supernatant containing milk oligosaccharides (MOs) was transferred to a fresh tube, frozen, and lyophilized. The protein pellet was kept for protein-linked glycan analysis.

Lyophilized milk oligosaccharides were resuspended in water at a concentration of 250 µL per mL of original milk. Residual proteins were eliminated using a spin-filter with a 10 kDa cutoff, operated at 11,000 rpm for 10 min (Sigma-Aldrich). Reduction for all glycan types was performed overnight at 50 °C using 0.5 M $NaBH_4$ and 20 mM NaOH. Desalting was accomplished using cation exchange resin (AG50WX8, Bio-Rad) packed onto a ZipTip C18 tip (Sigma-Aldrich). Following SpeedVac drying, methanol was introduced to remove any remaining borate through evaporation. The samples were subsequently resuspended in water at 250 µL per mL of original milk. The resulting glycans were analyzed via LC-MS/MS (see below) using 3 µL per measurement, with or without additional fractionation.

Further fractionation of the released MOs into neutral and acidic components was achieved using DEAE Sephadex A-25 (GH Healthcare). Briefly, 0.4 mL of conditioned DEAE Sephadex slurry was loaded into a 1 mL pipette tip packed with cotton wool. The column was washed three times with 0.5 mL Milli-Q water. Subsequently, the mixture containing the MOs was applied to the column. The flow-through, along with two additional 0.5 mL washes with Milli-Q water, contained the neutral MOs. Negatively charged MOs were then eluted with two successive 0.5 mL volumes of 1 M pyridine acetate (pH 5.4). After drying, residual pyridine was removed using a PGC cartridge, as described below. Since lactose, the predominant MO in the neutral fraction, would suppress other minor neutral MO signals during LC-MS/MS, it was eliminated using a carbon solid-phase cartridge (HyperSep Hypercarb SPE cartridges 25 mL, Thermo Scientific, Sweden). The cartridge was prepared with three 500 µL washes of 90% MeCN containing 0.1% TFA, followed by three 500 µL washes of 0.1% TFA. After MO application, lactose was eluted using three 500 µL washes of 8% MeCN with 0.1% TFA. The neutral MOs were subsequently eluted using three 500 µL washes of 65% MeCN containing 0.1% TFA, dried via centrifugation evaporation, and stored at −20 °C until analysis, maintaining a concentration of 250 µL per ml of original milk.

### Glycomics
LC-MS/MS analysis of all glycan types was performed from stock solutions containing 250 µL sample per mL of original milk. Glycans, at 3 µL per sample in water, underwent separation on an in-house packed column (10 cm × 250 µm) containing 5 µm porous graphitized carbon particles (Hypercarb; Thermo-Hypersil). Following injection, elution used an acetonitrile gradient (Buffer A: 10 mM ammonium bicarbonate; Buffer B: 10 mM ammonium bicarbonate in 80% acetonitrile). The gradient, ranging from 0 to 45% Buffer B, ran for 46 min, followed by a 100% Buffer B wash step and 24-min Buffer A equilibration.

Sample analysis was conducted in negative ion mode using an LTQ linear ion trap mass spectrometer (Thermo Electron), equipped with an IonMax standard ESI source featuring a stainless-steel needle maintained at −3.5 kV. Compressed air served as the nebulizer gas, while the heated capillary was maintained at 270 °C with a −50 kV capillary voltage. The process included a full scan ($m/z$ 340 or 380–2000, two micro scans, maximum 100 ms, target value of 30,000) followed by data-dependent $MS^2$ scans (two micro scans, maximum 100 ms, target value of 10,000) using normalized collision energy of 35%, an isolation window of 2.5 units, activation $q = 0.25$, and 30 ms activation time. The $MS^2$ threshold was set at 300 counts. Data acquisition and processing employed Xcalibur software (Version 2.0.7). Comparison of glycan abundances between samples involved

quantifying individual glycan structures relative to total content by integrating the extracted ion chromatogram peak area. The area under the curve (AUC) for each structure was normalized to the total AUC and expressed as a percentage. Peak area processing was performed using Progenesis QI (Nonlinear Dynamics Ltd). MOs were identified from their MS/MS spectra by manual annotation together with exoglycosidase verification as described previously[5,67,68].

For structural annotation, some assumptions were used in this study: elongation was assumed to occur from the lactose core structure; GalNAc was used for HexNAc when identified within a LacdiNAc sequence, otherwise, HexNAc was assumed to be GlcNAc; hexoses were interpreted as Gal residues, except for the Glc residue in lactose core. Elongation was assumed to occur via N-acetyl-lactosamine units (LacNAc or Galβ1-4GlcNAcβ1-3). Terminal epitopes corresponding to blood group ABH, type 1 and 2 chain, Lewis A/X, Lewis B/Y, αGal, and LacdiNAc were identified based on the sequences detected in their MS/MS spectra and sensitivity to different glycosidases. Neu5Ac linkage was identified using neuraminidase S (α2,3-specific, NEB, EC number 3.2.1.18), as well as the $^{0,2}X$ cleavage of Neu5Ac indicating α2,6-linkage[67]. The type 2 chain (Galβ1-4GlcNAc) was determined by β1,4-galactosidase S (NEB, EC number 3.2.1.23), and the cross-ring cleavage ($^{0,2}A_{GlcNAc}$-H$_2$O) of GlcNAc residues confirmed its presence[67]. The α1,3/6-galactosidase (NEB, EC number: 3.2.1.23) and α-N-acetylgalactosaminidase (NEB, EC number: 3.2.1.51) were used to exclude the potential for α-linkages. For the branching structure, the C/Z cleavage of branching units and $^{0,4}A$ cleavage of branching Gal residues were mainly used for characterization. Detected glycans and where to find them for the neutral and acidic fractions are recorded in Supplementary Data 12–13.

### Permethylation and MS data acquisition and processing
One of the MO samples (JC_231115MA5; seal B, day 7, acidic fraction) was permethylated and then fractionated by a Waters® OASIS MAX cartridge into neutral, mono-sulfated, and multiply sulfated glycan pools, exactly as described before[69]. Each of the permethylated MO sample fractions was initially screened by MALDI-MS to obtain an overall profile and to identify the major structures present by glycan compositions. Sample aliquots were mixed 1:1 with matrix (2,5-dihydrobenzyonic acid for positive mode analysis of non-sulfated glycans, and 10 mg/mL of 3,4-diaminobenzophenone for sulfated glycans in negative mode) in 50% acetonitrile and spotted onto the MALDI plate for data acquisition on an AB SCIEX MALDI TOF/TOF 5800 system. For nanoLC-nanoESI-MS analysis, the MO samples were dissolved in 10% acetonitrile/0.1% formic acid, applied via autosampler to an Ultimate™ 3000 RSLC system connected to an Orbitrap Fusion™ Tribrid™ Mass Spectrometer (ThermoFisher Scientific) via a PicoView nanosprayer (New Objective, Woburn, MA), and separated with a constant flow rate of 300 nL/min at 50˚C on a ReproSil-Pur 120 C18-AQ column (120 Å, 1.9 μm, 75 μm × 200 mm, Dr. Maisch). The solvent system used was buffer A (100% H$_2$O with 0.1% formic acid) and buffer B (100% acetonitrile with 0.1% formic acid), with a 60 min linear gradient of 30% to 80% B. The Orbitrap Fusion Tribrid instrument settings were as described previously[70] using a HCD-MS$^2$-product dependent MS$^3$ data dependent acquisition method for positive mode, with full MS and HCD MS$^2$ (stepped collision energy at 10, 15, 20) acquired in the Orbitrap at 120,000 and 30,000 resolution, respectively, and CID MS$^3$ (30% normalized collision energy) in the ion trap for the targeted MS$^2$ product ions at m/z 638.3382, 825.4227, and 999.5119, corresponding to Fuc$_1$Hex$_1$HexNAc$_1^+$, NeuAc$_1$Hex$_1$HexNAc$_1^+$, and NeuAc$_1$Fuc$_1$Hex$_1$HexNAc$_1^+$, respectively. For analysis of sulfated MO in negative mode, the data-dependent HCD-MS$^2$ were acquired with a stepped collision energy at 45, 55, 65. All MS and MS/MS data were manually assigned according to previously established fragmentation patterns[69–71].

### Data analysis
All data preprocessing and motif analyzes were performed using the Python package glycowork[23] (version 1.5). Analyzes using relative abundances all followed the same preprocessing workflow, which corresponds to the *preprocess_data* function in glycowork: outlier datapoints were removed via Winsorization and missing data were imputed via a MissForest-based, machine learning-driven imputation strategy. Then, data were transformed via a center-log ratio (CLR) transform, using a γ value of 0.1, to correct for the compositional data nature of glycomics data[41]. Glycan motifs were annotated via glycowork and motif abundances were obtained via the *quantify_motifs* function in glycowork. Our standard feature set here included "known" (named literature motifs) and "size_branch" (sizes and branching level of glycans), with any deviations from this noted in the respective figure legend.

### Synthesis of LacdiNAc-containing milk oligosaccharides
The total synthesis of LdiNnT from lactose was carried out through a sequential 13-step strategy involving glycosylation, deprotection, and purification. Full details on the experimental procedures, physical data, and $^1$H, $^{13}$C, $^1$H-$^1$H COSY, and $^1$H-$^{13}$C HSQC NMR spectra of all compounds are provided in the Supporting Information (SI), in the section Supplementary Methods.

### Immunomodulation of polarized macrophages
THP−1 cells (obtained from ATCC, TIB-202; CVCL_0006) were cultivated in Roswell Park Memorial Institute 1640 medium (RPMI 1640; Gibco, A1049101) supplemented with 10% (v/v) fetal bovine serum (FBS; Nordic Biolabs, FBS-HI−12A) and 1% (v/v) penicillin-streptomycin (Sigma-Aldrich, P4333−100ML) in a humidified atmosphere containing 5% CO$_2$ at 37 °C. $4 \times 10^4$ cells/well were seeded in a 96-well plate and differentiated into naïve macrophage-like cells (M0 macrophages) by treatment with 25 nM phorbol-12-myristate-13-acetate (PMA; Sigma-Aldrich, P8139-1MG) for 48 h. A successful differentiation reaction indicated the status of the provided cells as monocytes. The cells were not otherwise authenticated. After an additional 24-h rest period in fresh medium, the cells were challenged with 100 ng/mL lipopolysaccharide (LPS; Sigma-Aldrich, L4391-1MG) (M1-polarized macrophages) or 20 ng/mL interleukin 4 (IL-4; Sigma-Aldrich, SRP3093-20UG) and 20 ng/mL interleukin 13 (IL-13; Sigma-Aldrich, SRP3274-10UG) (M2-polarized macrophages) in the absence or presence of 0.1–1 mM chemically pure MOs (N-acetyllactosamine: LacNAc, Sigma-Aldrich, A7791-5MG; N,N-acetyllactosamine: LacdiNAc, GlycoNZ, GNZ-0001-sp; lacto-N-neotetraose: LNnT, Elicityl, GLY021-95%; lacto-N,N-neotetraose: LdiNnT, synthesized in-house). After 24 h, the culture supernatants were collected and analyzed for the levels of key macrophage cytokines using a multiplex immunoassay based on fluorescence-encoded beads (LEGENDplex; Biolegend, 740503) according to the manufacturer's instructions. The samples from three biological replicates were then measured on an Accuri C6 Plus Flow Cytometer (BD) and the results were analyzed using LEGENDplex Data Analysis Software Suite (Biolegend, Qognit).

### Bacterial strains and crystal violet assay for biofilm quantification
S. aureus CCUG 1800T and S. agalactiae CCUG 4208T were obtained from the Culture Collection University of Gothenburg (CCUG, https://www.ccug.se/). K. pneumoniae KP1 were obtained from Dr. Scott Rice (University of Technology, Sydney, Australia). K. pneumoniae KP1 and S. aureus CCUG 1800T were maintained in brain-heart infusion broth (Sigma-Aldrich, 53286-500G) and on brain-heart infusion agar. S. agalactiae CCUG 4208T were maintained in tryptic-soy broth (Sigma-Aldrich, 22092-500G) and on tryptic soy agar. All strains were preserved as glycerol stocks at −80 °C.

The impact on biofilm formation by milk oligosaccharides was assessed by crystal violet staining. Overnight cultures from single colonies were diluted to a final $OD_{600}$ of 0.01 (*K. pneumoniae* and *S. aureus*) and 0.1 (*S. agalactiae*) in a 96-well plate with a total well volume of 100 μL. Milk oligosaccharides (*N*-acetyllactosamine: LacNAc, Sigma-Aldrich, A7791-5MG; *N,N*-acetyllactosamine: LacdiNAc, GlycoNZ, GNZ-0001-PA; 2′-fucosyllactose: 2′-FL, Sigma-Aldrich, SMB00933-50MG; 6′-sialyl-d-lactose: 6′-SL, Sigma-Aldrich, 40817-1MG; D-glucuronic acid: GlcA, Sigma-Aldrich, G5269-10G; glucoronyllactose: GlcA-Lac, Elicityl) were added to a final concentration of 1 mg/mL. Plates were sealed with a breathable, sterile film and incubated statically for 24 h at 37 °C to allow for biofilm formation. After 24 h, $OD_{600}$ was measured using a Varioskan LUX Multimode Microplate reader. The planktonic cells were then removed, and each well was washed thrice with 150 μL 1×PBS. The remaining biomass was stained with a 0.1% crystal violet solution for 60 min at room temperature. After staining, the wells were washed once with 300 μL 1×PBS, followed by two washes with 150 μL 1×PBS. The stain was solubilized by adding 150 μL 33% acetic acid for 30 min and absorbance was read at 595 nm using a Varioskan LUX Multimode Microplate reader. The assay was performed once ($n = 1$) for LacNAc, LacdiNAc, and 6′-SL, twice ($n = 2$) for GlcA and GlcA-Lac, and thrice ($n = 3$) for 2′-FL, with three technical replicates for each condition. All values were corrected by subtracting the values of the blanks, and the mean values was used for plotting.

### Statistical analysis
All statistical testing has been done in Python 3.12.6 using the glyco-work package (version 1.5), the statsmodels package (version 0.14) and the scipy package (version 1.11). Data normalization and motif quantification was done with glycowork (version 1.5). All statistical operations on glycomics and metabolomics data have been performed on CLR-transformed data. Testing differences between two groups used Welch's t-test, while testing differences between more than two groups used ANOVA or ANOSIM, followed by Tukey's Honestly Significant Difference post-hoc tests in the former case. Regressions were performed via Spearman correlation analyzes of the data, or of the residuals in the case of regularized partial correlations. All multiple testing correction has been performed via a two-stage Benjamini–Hochberg correction.

### Ethics statement
Work involving animals in this study complied with all relevant ethical regulations and was licensed under the UK Home Office project 60/4009 or preceding versions and conformed to the UK Animals (Scientific Procedures) Act, 1986. Research was approved by the University of St Andrews Animal Welfare and Ethics Committee.

### Reporting summary
Further information on research design is available in the Nature Portfolio Reporting Summary linked to this article.

## Data availability
Unless otherwise stated, all data supporting the results of this study can be found in the article, supplementary, and source data files. All processed data generated and/or used in this article can either be found in supplemental tables or as stored datasets within glycowork [https://github.com/BojarLab/glycowork][23]. The glycomics MS raw files have been deposited in the GlycoPOST database under the ID GPST000556, with the corresponding retention times and compositions found in Supplementary Data 12-13. Source data are provided with this paper.

## Code availability
Code and documentation are available via GitHub [https://github.com/BojarLab/glycowork][72].

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

## Acknowledgements

The authors would like to thank James Urban for facilitating sample acquisition and transport. This work was supported by a Branco Weiss Fellowship – Society in Science awarded to D.B.; by the Knut and Alice Wallenberg Foundation; the Hasselblad Foundation; and the University of Gothenburg, Sweden. C.C. and R.H. gratefully acknowledge support from the Swiss National Science Foundation (project 320030-231409) and the University of Basel, Switzerland. We thank SciLifeLab and BioMS (Swedish research council) for providing financial support to the Proteomics Core Facility, Sahlgrenska Academy. K-H.K. was supported by Academia Sinica grant AS-IR-113-L04. We thank the Academia Sinica Common Mass Spectrometry Facilities for Proteomics and Protein Modification Analysis funded by the Academia Sinica Core Facility and Innovative Instrument Project grant AS-CFII-108-107, for MS data collection. Sample collection from mother seals on the Isle of May was funded from core support given to the Sea Mammal Research Unit, Scottish Oceans Institute, from the Natural Environmental Research Council (United Kingdom). The funders had no role in study design, data collection and analysis, decision to publish or preparation of the manuscript.

## Author contributions

D.B. conceptualization; A.R.B., D.B., and J.L. formal analysis; C.C. and R.H. synthesis design & NMR analysis; C.C., D.B., J.L., M.W.K., P.P.P., and R.H. resources; D.B. data curation; A.R.B., D.B., C.J., and J.L. writing–original draft; A.R.B., C.C., D.B., C.J., J.B.-P., J.L., K.-H.K., M.D., R.H., M.W.K., and S.-Y.G writing–review & editing; A.R.B., C.J., D. B., J.L., K.-H.K., and S.-Y.G. visualization; D.B., J.B.-P., K.-H.K., and R.H. supervision; D.B. funding acquisition; C.J., D.B., J.L., K.-H.K., M.D., and S.-Y.G. methodology; C.J., J.L., K.-H.K., and M.D. validation. J.L. conducted all cytokine multiplex analyses. M.D. conducted all biofilm experiments. C.J., K.-H.K., and S.-Y.G. performed all glycomics measurements. A.R.B. and D.B. wrote the code for analyzing the systems biology data. M.W.K. and P.P.P. performed sample collection.

## Funding

## Competing interests

D.B. is consulting on glycobiology-related topics via SweetSense Analytics AB. The remaining authors declare no competing interests.
