## [Transparent Peer Review file · Nature Communications]

Seal milk oligosaccharides rival human milk complexity and exhibit functional dynamics during lactation

Corresponding Author: Dr Daniel Bojar

Version 0:

Reviewer comments:

Reviewer #1

(Remarks to the Author)

The paper by Jin, et al., describes the full glycomics analysis of milk oligosaccharides (MO) from the Atlantic grey seal *Halichoerus grypus* (*H. grypus*). The authors covered the changes in MO through the lactation period of the seal and found the complexity and the changes rivaled those in human MO. This study is quite comprehensive, insofar as over 300 MO have been characterized and quantitated, while discovering 166 novel structures, as is claimed by the authors. I feel *Nat. Comm.* is an appropriate vehicle for this study as the data seems complete and is described in acceptable detail. The comparison with metabolomic data coupled with biological/immunological activity profiling of several new structures adds to the overall complexity and completeness of the study. I enthusiastically recommend publication after some minor technical and data details are explained; these are listed below:

1) As in many glycomics or glycoproteomic studies, linkage stereochemistry is probably the most difficult structural feature to characterize. How confident are the authors that all the α and β linkages are accurate based on the methods used? I ask since there are nearly 200 new structures identified. Are there any methods to corroborate these by comparison to similar oligosaccharides? I know the authors synthesized the LacdiNAc tetrasaccharide from scratch (which was commendable!) But just curious of the "confidence" that you have for all new structures.

2) What is also commendable is the synthetic details and the NMR characterization of all new molecules. The authors seemed to take great pains to identify as many protons and carbons as possible and it is always satisfying to see full characterization data (actually assigning the peaks), as opposed to "7.5, d, J = 8 Hz, 1 H". Regarding this, for compound 12, I found it fascinating that near every benzyl methylene group was diastereotopic: i.e., two doublets per set, instead of singlets centered around 5.2-5.4 ppm. Even more interesting were the COSY cross peaks from aromatic residues to peaks in the 4-5 ppm range, suggesting long-range (4-bond) coupling to the aromatic protons. This would mean there should be "doublets of doublets" for some benzyl CH₂ protons. This is not seen for compound 11, the trisaccharide that also contains several benzyl groups! Can the authors offer any explanation for this or have they seen this in other structures? It would be interesting to do a full conformational analysis of compound 12.

3). The discovery of specific LacdiNAc structures that are novel and also have relevant biological activity is a very interesting find. LacdiNAc has been found on tumor cells and is known to interact with Galectin-3. LacNAc is considered the endogenous glycan ligand for Gal-3; the authors correctly point out that another receptor is most likely being targeted since LacNAc does show the same activity as LacdiNAc. Since the authors have prepared a LdiNnT, a lacto-N,N-neotetraose derivative, this could be used to test galectin binding on either lectin arrays or attach LdiNnT to a glycan microarray and test lectin binding. Since this acts on M2-polarize macrophages, have the authors examined which lectins or lectin-like proteins are exposed on these cells that may interact with LacdiNAc structures? It is difficult to actually discern the differences in the activity between the LacdiNAc di- and tetrasaccharide structures in this section of the manuscript. Additional comparisons/explanations in the text would help. Please enhance the X-axis labels in Figure 5, they are barely discernable. Lastly, describing LdiNnT as simply "lacto-N,N-neotetraose" (basically lacto-N-neotetraose with an extra "N") to me is confusing, since it is no longer "lacto"--it is "LacdiN". A better name would help the reader differentiate the two structures.

4) As has been shown for mucin-type glycans and also HMOs, mixtures of MO structures can often have enhanced or novel activity that individual components do not possess on their own. Have the authors considered mixing specific structures in a systematic way, perhaps based on literature and explore various activities?

5) Minor point as first mentioned above: The labels on the axes of Figures 5, and S8 should be expanded and bolded for better visualization

I think this is a well-designed and performed study that will enhance the field and foster new research into MO structures and function. It is comprehensive and all the necessary data is included and available. I think it should be published after minor revisions.

Reviewer #2

(Remarks to the Author)

In this study, the authors characterized in great detail the chemical structures of a huge variety of oligosaccharides, including very complicated structures in the milk of grey seal. The characterized structures include the novel ones, which have not been identified in any mammalian milk. As they developed the method for characterization of oligosaccharide structures including minor ones using a small scale of the samples, I agree that significance is recognized with respect of advanced glycomics.

If one think that the study focused on the structural study of milk oligosaccharides for only one species, I'm afraid that this might be underestimated. However, the lineage evolution is discussed in detail to compare the milk oligosaccharide structures between the grey seal and other mammalian species including humans. I believe that the explore of the evolution of milk oligosaccharides is one area of the evolutionary biology. Some review papers have been published for the subject of evolution of milk oligosaccharides and lactose. I rate highly the scientific significance of the paper, as this study deepened and developed the story of this evolution.

The authors compared the complicated milk oligosaccharide structures of grey seal with those of human milk oligosaccharides, which have been studied in many studies for a long time. Human female gives birth of relatively altricial newborns, whereas seal female gives birth of the precocial newborns. The lactation period is much longer in human than that in seals. Although it is common between grey seal and human with respect that both have very complicated structures of milk oligosaccharides, I think that there is a difference in the physiological significance of both milk oligosaccharides, because the concentration is much higher in human than in grey seal.

As seal milks contain extremely low concentration of carbohydrates, the neonates must prefer lipids to lactose as their nutritional source. It is believed that the decrease of the concentration of lactose should be caused by decrease of the expression level of α -lactalbumin in lactating mammary glands. This means that the biosynthetic rate of lactose is slow. When this rate is slow, the relative biosynthetic rate of milk oligosaccharides will be enhanced, as the rate of biosynthesis of lactose would not be a rate-limiting stage for the glycosyltransferases, which transfer monosaccharides to lactose, to catalyze the biosynthesis of milk oligosaccharides. As a result, a huge of the complicated milk oligosaccharides may occur in seal milk. Namely, the existence of the complicated milk oligosaccharides has been gained by accidently rather than as the result of positive selective advantage of the biological significance. Human milk contains around 11 gram/L of milk oligosaccharides, which is high concentration in milk. One may think that the milk oligosaccharides are not so significant in seal compared in human. I hope that the authors will have the answer for this thought.

In this study, the authors performed the organic synthesis of LdiNnT, which contains LacdiNAc unit, and studied to explorer its biological functions as immunomodulatory and anti-biofilm property in addition to the structural characterization of grey seal milk oligosaccharides. Although I understand that it is significant to explorer the biological functions of grey seal specific milk oligosaccharides, it seems that the focus of the paper is rather dispersed if the organic synthesis of LdiNAc and explorer the functions are included in a paper with the structural study of grey seal milk oligosaccharides. I want to ask for the editor's judgement whether the organic synthesis of LdiNnT and the explorer of the biological functions would be included in this paper. I will follow the judgement by the editor in this point.

I suggest that the authors will add some descriptions in Discussion for the following issue.

1. In hooded seal milk oligosaccharides, sulphate linked at OH-3 of non-reducing Gal (ref. 13, 14). I suggest that the difference of the linked positions of sulphate in milk oligosaccharides between grey and hooded seals will be described.
2. The oligosaccharides containing poly LacNAc were identified not only in seals' milks but also in raccoon milk (ref. 26). I suggest that the authors will expand the discussion on the significance of poly LacNAc in other carnivora milk.
3. The oligosaccharides containing poly LacNAc were also found in the milks of hooded and bearded seals (ref. 14). Are there structural difference of the poly LacNAc containing milk oligosaccharides among three species of seals?
4. It is likely that grey seal milk contains only type 2 oligosaccharides; this differs from human milk, which contains the predominant type 1 oligosaccharides as well as the type 2. I suggest that they discuss the physiological reason of this difference between both species. Please refer ref 25.
5. It is likely that grey seal milk did not contain Gal β 1-3(Gal β 1-4GlcNAc β 1-6)Gal β 1-4Glc (novo LNP 1), which have been identified in milk/colostrum of cows, goats, sheep, camels, pigs and horses. I suggest that they will describe it.

Comments for individual parts

1. Line 20 and 37

I suggest that “, especially in human” will be added after “and health” in line 20 and 37. In line 37 ~ 43, the authors described the physiological significances of milk oligosaccharides, which have been shown by many studies. However, these functions are shown for human milk oligosaccharides and related to the health of human infants.

Although such biological significance has been clarified for human milk oligosaccharides, the biological functions should be also clarified for milk oligosaccharides of other species.

2. Line 67 and 68

I suggest that “phocid” will be added before “seal species” in line 67 and Australian fur seal in line 68 will be removed. As fur seal milk does not contain milk oligosaccharides as well as lactose, this status must be different from the milks of phocid seal.

3. Line 144 ~ 146

I suggest that the authors will add the description that giants MOs were found in the milks of hooded and bearded seals, too, as in ref. 14.

4. Line 157~158

It is the first finding that oligosaccharides containing sialyl-Lewis x, Lewis y, B antigen, or Galili antigen are contained in any seal milks. The milk oligosaccharides, containing Lewis x, B antigen or Galili antigen, were identified in some carnivora species (ref. 57, 58). However, the milk oligosaccharides, containing A antigen, were identified in some carnivora species, whereas that was not detected in the grey seal sample. I suggest that the authors will add this short description in Discussion.

5. Line 165~168

Although the existence of the unusual oligosaccharides, containing Neu5Ac α 2-6 residue, were suggested in seal milks in the previous studies (ref. 12, 13), the linked position of Neu5Ac α 2-6 was not completely determined by the characterization using NMR. It must be significant that the authors could determine this position with their advanced technique. This is just my thoughts.

6. After line 246

It can be evaluated that they found the change of milk oligosaccharide profiling of grey seal during the course of lactation, despite that the lactation period is short.

7. Line 305~307

They found that the concentration of Fuc α 1-2Gal-containing MOs decreases, while that of Fuc α 1-3Glc(NAc)-containing MOs increases during the course of lactation. This status is similar to that in human.

8. Line 518

The condition of ion exchange chromatography with DEAE Sephadex A-25 must be described. How was the volume of the resin in the column? How was the eluted condition of acidic MOs, which had been adsorbed in the resin?

Version 1:

Reviewer comments:

Reviewer #1

(Remarks to the Author)

The authors have answered all the queries of both reviewers to (at least) my satisfaction, and hence I feel the manuscript is now ready for publication--nice work!

Reviewer #2

(Remarks to the Author)

As the paper has been revised well to answer for the comments from me, I agree that the paper now acceptable for publication in Nature Communications.

We thank both reviewers for their insightful comments and suggestions for improvement. We have fully addressed these comments in our substantially revised manuscript by engaging in extensive text modifications and additions, as well as visual improvement of our figures, where indicated. In summary, we have (i) added a new figure to further document our structure assignments (**new Fig. S2**), (ii) expanded our methodology section to make structure assignment and workflows more transparent, (iii) expanded and contextualized our text, including appropriate references, to make the implications of our findings clearer, and (iv) improved the readability of our figures where needed. Changes in the manuscript and point-by-point responses here are colored in blue. We believe that these changes have substantially improved our manuscript, contextualized our findings, and will allow readers to better evaluate our analyses and findings.

Reviewer #1 (Remarks to the Author):

The paper by Jin, et al., describes the full glycomics analysis of milk oligosaccharides (MO) from the Atlantic grey seal *Halichoerus grypus* (*H. grypus*). The authors covered the changes in MO through the lactation period of the seal and found the complexity and the changes rivaled those in human MO. This study is quite comprehensive, insofar as over 300 MO have been characterized and quantitated, while discovering 166 novel structures, as is claimed by the authors. I feel *Nat. Comm.* is an appropriate vehicle for this study as the data seems complete and is described in acceptable detail. The comparison with metabolomic data coupled with biological/immunological activity profiling of several new structures adds to the overall complexity and completeness of the study. I enthusiastically recommend publication after some minor technical and data details are explained; these are listed below:

We thank the reviewer for their assessment of our work and respond to individual comments below.

1) As in many glycomics or glycoproteomic studies, linkage stereochemistry is probably the most difficult structural feature to characterize. How confident are the authors that all the α and β linkages are accurate based on the methods used? I ask since there are nearly 200 new structures identified. Are there any methods to corroborate these by comparison to similar oligosaccharides? I know the authors synthesized the LacdiNAc tetrasaccharide from scratch (which was commendable!) But just curious of the “confidence” that you have for all new structures.

We agree with the reviewer that this is a challenging task and we have now expanded our description of our workflow to elucidate these structural details in the revised manuscript, especially in the revised Methods section. Briefly, we used a combination of established diagnostic ions for certain linkages, exoglycosidase/sulfatase digestion for other linkages/modifications, and, if available, comparison to existing spectra from milk oligosaccharides from other species that have been previously measured by us (Jin et al., *Mol Cell Proteomics*, 9:100635, 2023). For linkages for which we did not have sufficient evidence,

we denoted wildcards in the glycan sequence, such as “GalOS(b1-4)Glc-ol” for an undefined sulfate position, or “Neu5Ac(a2-?)Gal(b1-4)GlcNAc(b1-3)Gal(b1-4)GlcNAc(b1-6)[Neu5Ac(a2-?)Gal(b1-4)GlcNAc(b1-3)]Gal(b1-4)Glc-ol” for ambiguous sialic acid linkages. In the course of this revision, we have also added another figure to document some of our assignments, focusing on sialic acid linkages, with the **new Figure S2**.

2) What is also commendable is the synthetic details and the NMR characterization of all new molecules. The authors seemed to take great pains to identify as many protons and carbons as possible and it is always satisfying to see full characterization data (actually assigning the peaks), as opposed to “7.5, d, J = 8 Hz, 1 H”. Regarding this, for compound 12, I found it fascinating that near every benzyl methylene group was diastereotopic: i.e., two doublets per set, instead of singlets centered around 5.2-5.4 ppm. Even more interesting were the COSY cross peaks from aromatic residues to peaks in the 4-5 ppm range, suggesting long-range (4-bond) coupling to the aromatic protons. This would mean there should be “doublets of doublets” for some benzyl CH₂ protons. This is not seen for compound 11, the trisaccharide that also contains several benzyl groups! Can the authors offer any explanation for this or have they seen this in other structures? It would be interesting to do a full conformational analysis of compound 12.

We thank the reviewer for their thorough assessment of our NMR characterization. Regarding the diastereotopic nature of the benzylic methylene groups, this is in fact a rather common phenomenon for benzylic protecting groups on glycans, perhaps due to the chiral richness of glycan structures. This diastereotopic nature of benzyl protecting groups has been reported by both us (doi: 10.1002/hlca.202400026, 10.1002/open.202200134) and others (doi: 10.1039/d0ob00172d, 10.1002/celc.201900215, 10.1039/C3RA45658G) across diverse synthetic targets.

Regarding the long-range coupling observed between benzylic protons and aryl protons in the COSY spectrum of compound 12, this is indeed a rather interesting observation. Upon closer inspection, this long-range coupling is present to varying degrees in all of the Bn-containing structures in this publication, but was only visible in this particular spectrum of the SI due to the signal intensity for this spectrum being set to a higher baseline intensity than all the other compound spectra. To improve visual consistency in our SI images, we have replaced the COSY spectrum for compound 12 with the same spectrum but at a lower baseline intensity, so that it more closely aligns with NMR spectra for the other compounds.

Interestingly, looking into this phenomenon further, we see this long-range coupling not only in the Bn-containing intermediates in this manuscript, but also in non-related aryl-containing structures across diverse projects, suggesting that this may be a relatively wide-spread phenomenon. The long-range COSY correlations are relatively weak, and therefore only visible when the signal intensity is significantly increased. Given this weak intensity, we anticipate that the coupling constant would be <1 Hz and this would explain why (with a 500 MHz NMR instrument) the ¹H NMR signals do not appear as “doublet of doublets.”

3). The discovery of specific LacdiNAc structures that are novel and also have relevant biological activity is a very interesting find. LacdiNAc has been found on tumor cells and is known to interact with Galectin-3. LacNAc is considered the endogenous glycan ligand for Gal-3; the authors correctly point out that another receptor is most likely being targeted since LacNAc does show the same activity as LacdiNAc. Since the authors have prepared a LdiNnT, a lacto-N,N-neotetraose derivative, this could be used to test galectin binding on either lectin arrays or attach LdiNnT to a glycan microarray and test lectin binding. Since this acts on M2-polarize macrophages, have the authors examined which lectins or lectin-like proteins are exposed on these cells that may interact with LacdiNAc structures?

We share the reviewer's enthusiasm for identifying the individual receptors that mediate the differential effects of distinct yet closely related structures, and we aim to address this and other related questions (e.g., point 4 below) in future research, as it will require the chemical synthesis of additional milk oligosaccharides.

Macrophage Galactose-type Lectin (MGL) is highly expressed in macrophages and is further upregulated upon alternative (i.e., M2) activation. In humans, MGL specifically recognizes GalNAc-terminated structures, including Tn and LacdiNAc antigens in tumor cell lines, but not Gal-terminated structures. It could therefore be a candidate receptor that explains the differential effects of LacNAc- versus LacdiNAc-containing MOs observed in our experiments. Further studies will, of course, be required to investigate this possibility. We have added this point to the revised Discussion in the manuscript.

References: [10.1016/j.it.2007.10.010](https://doi.org/10.1016/j.it.2007.10.010). DOI: [10.1016/j.bbagen.2020.129513](https://doi.org/10.1016/j.bbagen.2020.129513)

It is difficult to actually discern the differences in the activity between the LacdiNAc di- and tetrasaccharide structures in this section of the manuscript. Additional comparisons/explanations in the text would help. Please enhance the X-axis labels in Figure 5, they are barely discernable. Lastly, describing LdiNnT as simply "lacto-N,N-neotetraose" (basically lacto-N-neotetraose with an extra "N") to me is confusing, since it is no longer "lacto"--it is "LacdiN". A better name would help the reader differentiate the two structures.

For naming our LacdiNAc-containing milk oligosaccharides, we have based our naming here on two precedents: (i) the initial naming in our previous work, in which we initially identified this type of structure in various mammals (doi: [10.1016/j.mcpro.2023.100635](https://doi.org/10.1016/j.mcpro.2023.100635)) and (ii) to indicate the relationship to the typical/expected substructure in this position, especially as the traditional lacto-N-neotetraose already no longer presents a lacto-substructure (since lacto-N-biose is basically a LacNAc unit). We thus believe maintaining such a continuity with established milk oligosaccharide nomenclature will allow readers to more easily situate and contextualize these new structures.

4) As has been shown for mucin-type glycans and also HMOs, mixtures of MO structures can often have enhanced or novel activity that individual components do not possess on their own. Have the authors considered mixing specific structures in a systematic way, perhaps based on literature and explore various activities?

Complex mixtures of MOs have indeed been shown to exert enhanced and/or differential effects in comparison to the individual components in isolation, suggesting the presence of complex, interdependent regulatory mechanisms that warrant further investigation. However, more systematic studies expanding beyond the limited catalogue of commercially available MOs remain technically challenging due to the prohibitively expensive efforts required for chemical synthesis of large libraries of complex structures. We have included this point in our revised Discussion.

5) Minor point as first mentioned above: The labels on the axes of Figures 5, and S8 should be expanded and bolded for better visualization

We agree with the reviewer and have improved the axis labels accordingly.

I think this is a well-designed and performed study that will enhance the field and foster new research into MO structures and function. It is comprehensive and all the necessary data is included and available. I think it should be published after minor revisions.

We thank the reviewer for their thorough assessment and we have improved our manuscript according to the comments above.

Reviewer #2 (Remarks to the Author):

In this study, the authors characterized in great detail the chemical structures of a huge variety of oligosaccharides, including very complicated structures in the milk of grey seal. The characterized structures include the novel ones, which have not been identified in any mammalian milk. As they developed the method for characterization of oligosaccharide structures including minor ones using a small scale of the samples, I agree that significance is recognized with respect of advanced glycomics.

We are grateful for the feedback on our manuscript by the reviewer, which has allowed us to improve our work during the revision. Our answers to individual comments can be found below.

If one think that the study focused on the structural study of milk oligosaccharides for only one species, I'm afraid that this might be underestimated. However, the lineage evolution is discussed in detail to compare the milk oligosaccharide structures between the grey seal and other mammalian species including humans. I believe that the explore of the evolution of milk oligosaccharides is one area of the evolutionary biology. Some review papers have been published for the subject of evolution of milk oligosaccharides and lactose. I rate highly the scientific significance of the paper, as this study deepened and developed the story of this evolution.

The authors compared the complicated milk oligosaccharide structures of grey seal with those of human milk oligosaccharides, which have been studied in many studies for a long time. Human female gives birth of relatively altricial newborns, whereas seal female gives birth of the precocial newborns. The lactation period is much longer in human than that in seals. Although it is common between grey seal and human with respect that both have very complicated structures of milk oligosaccharides, I think that there is a difference in the physiological significance of both milk oligosaccharides, because the concentration is much higher in human than in grey seal.

As seal milks contain extremely low concentration of carbohydrates, the neonates must prefer lipids to lactose as their nutritional source. It is believed that the decrease of the concentration of lactose should be caused by decrease of the expression level of α -lactalbumin in lactating mammary glands. This means that the biosynthetic rate of lactose is slow. When this rate is slow, the relative biosynthetic rate of milk oligosaccharides will be enhanced, as the rate of biosynthesis of lactose would not be a rate-limiting stage for the glycosyltransferases, which transfer monosaccharides to lactose, to catalyze the biosynthesis of milk oligosaccharides. As a result, a huge of the complicated milk oligosaccharides may occur in seal milk. Namely, the existence of the complicated milk oligosaccharides has been gained by accidently rather than as the result of positive selective advantage of the biological significance. Human milk contains around 11 gram/L of milk oligosaccharides, which is high concentration in milk. One may think that the milk oligosaccharides are not so significant in seal compared in human. I hope that the authors will have the answer for this thought.

We agree with the reviewer that milk oligosaccharides may serve different functional roles in humans and in seals, and now have added this information to our revised Introduction. While we were unable to determine the absolute concentration of milk oligosaccharides in our milk samples, we are confident in stating that the total amount of seal MOs is lower than in human milk and now mention this in our revised Discussion.

In this study, the authors performed the organic synthesis of LdiNnT, which contains LacdiNAC unit, and studied to explorer its biological functions as immunomodulatory and anti-biofilm property in addition to the structural characterization of grey seal milk oligosaccharides. Although I understand that it is significant to explorer the biological functions of grey seal specific milk oligosaccharides, it seems that the focus of the paper is rather dispersed if the organic synthesis of LdiNAC and explorer the functions are included in a paper with the structural study of grey seal milk oligosaccharides. I want to ask for the editor's judgement whether the organic synthesis of LdiNnT and the explorer of the biological functions would be included in this paper. I will follow the judgement by the editor in this point.

We thank the reviewer for their assessment yet, after communication from the editor, have kept the synthesis and functional analysis of our LacdiNAC-containing milk oligosaccharides in this paper, to appeal to a broader audience and present a more holistic exploration of the seal milk glycome and its functions, including compounds with the potential for future biomedical relevance.

I suggest that the authors will add some descriptions in Discussion for the following issue.

1. In hooded seal milk oligosaccharides, sulphate linked at OH-3 of non-reducing Gal (ref. 13, 14). I suggest that the difference of the linked positions of sulphate in milk oligosaccharides between grey and hooded seals will be described.

We have added this information to page 7, where we describe the sulfated milk oligosaccharides in *H. grypus*.

2. The oligosaccharides containing poly LacNAc were identified not only in seals' milks but also in raccoon milk (ref. 26). I suggest that the authors will expand the discussion on the significance of poly LacNAc in other carnivora milk.

We thank the reviewer for pointing out this connection and, after carefully comparing the poly-LacNAc profiles in Carnivora milk, we have come to the conclusion that this seems to be a conserved element, as both the extension mode (predominantly an iterative extension of the GlcNAc β 1-6 branch) and the type 2 LacNAc units are maintained in the known cases. We have added this information to the revised Discussion.

3. The oligosaccharides containing poly LacNAc were also found in the milks of hooded and bearded seals (ref. 14). Are there structural difference of the poly LacNAc containing milk oligosaccharides among three species of seals?

We thank the reviewer for pointing this out. Our seal MOs, as those from other seals, are indeed especially rich in the branched I antigen, in contrast to the linear i antigen. The three main differences of the poly-LacNAc oligosaccharides across seal species lie in: (i) sulfation position (Gal3S in other species vs GlcNAc6S here), (ii) point-of-branching (predominantly GlcNAc β 1-3 in other species vs both GlcNAc β 1-3, and especially GlcNAc β 1-6 for the longest MOs, here), and (iii) type of decoration (Fuc in other species vs both Fuc and Neu5Ac here). We have added these considerations to the revised manuscript in the appropriate positions.

4. It is likely that grey seal milk contains only type 2 oligosaccharides; this differs from human milk, which contains the predominant type 1 oligosaccharides as well as the type 2. I suggest that they discuss the physiological reason of this difference between both species. Please refer ref 25.

We have now added this information (and reference) to the revised text where we discuss the dominance of type 2 oligosaccharides in grey seal milk.

5. It is likely that grey seal milk did not contain Gal β 1-3(Gal β 1-4GlcNAc β 1-6)Gal β 1-4Glc (novo LNP 1), which have been identified in milk/colostrum of cows, goats, sheep, camels, pigs and horses. I suggest that they will describe it.

We have added this information to the revised text where we describe the seal milk glycome, to also indicate what we do not detect (such as *novo*-LNP-I).

Comments for individual parts

1. Line 20 and 37

I suggest that “, especially in human” will be added after “and health” in line 20 and 37. In line 37 ~ 43, the authors described the physiological significances of milk oligosaccharides, which have been shown by many studies. However, these functions are shown for human milk oligosaccharides and related to the health of human infants.

Although such biological significance has been clarified for human milk oligosaccharides, the biological functions should be also clarified for milk oligosaccharides of other species.

We thank the reviewer for this suggestion and have now added information and references regarding the physiological roles of milk oligosaccharides in non-human mammals to the revised Introduction.

2. Line 67 and 68

I suggest that “phocid” will be added before “seal species” in line 67 and Australian fur seal in line 68 will be removed. As fur seal milk does not contain milk oligosaccharides as well as lactose, this status must be different from the milks of phocid seal.

We have amended our manuscript as suggested here.

3. Line 144 ~ 146

I suggest that the authors will add the description that giants MOs were found in the milks of hooded and bearded seals, too, as in ref. 14.

We have added this information to the revised manuscript and also reference the indicated paper now at this position.

4. Line 157~158

It is the first finding that oligosaccharides containing sialyl-Lewis x, Lewis y, B antigen, or Galili antigen are contained in any seal milks. The milk oligosaccharides, containing Lewis x, B antigen or Galili antigen, were identified in some carnivora species (ref. 57, 58). However, the milk oligosaccharides, containing A antigen, were identified in some carnivora species, whereas that was not detected in the grey seal sample. I suggest that the authors will add this short description in Discussion.

We agree with the reviewer and have expanded our comparison with Carnivora milk oligosaccharide motifs in the revised Discussion.

5. Line 165~168

Although the existence of the unusual oligosaccharides, containing Neu5Ac α 2-6 residue, were suggested in seal milks in the previous studies (ref. 12, 13), the linked position of Neu5Ac α 2-6 was not completely determined by the characterization using NMR. It must be significant that the authors could determine this position with their advanced technique. This is just my thoughts.

We agree with the reviewer that this substructure is remarkable and we have used our mass spectrometry-driven pipeline to ensure its accurate assignment (detailed in Fig. S1e, for instance). Especially the diagnostic $^{0,2}X_{\text{Neu5Ac}}$ fragments here allowed us to assign this linkage, combined with its positioning on the C3 branch, given the size of the observed D ions. We also routinely used α 2,3-specific sialidase to exclude the alternative linkage and now additionally showcase this, as well as the diagnostic ions, in the **new Figure S2**.

6. After line 246

It can be evaluated that they found the change of milk oligosaccharide profiling of grey seal during the course of lactation, despite that the lactation period is short.

We thank the reviewer for their observation and have added this crucial information to the revised manuscript.

7. Line305~307

They found that the concentration of Fuca α 1-2Gal-containing MOs decreases, while that of Fuca α 1-3Glc(NAc)-containing MOs increases during the course of lactation. This status is similar to that in human.

We have included this information in the revised manuscript.

8. Line 518

The condition of ion exchange chromatography with DEAE Sephadex A-25 must be described. How was the volume of the resin in the column? How was the eluted condition of acidic MOs, which had been adsorbed in the resin?

We have added this information to the revised Methods section.